# In-context learning in presence of spurious correlations

## Abstract

Large language models exhibit a remarkable capacity for in-context learning, where they learn to solve tasks given a few examples. Recent work has shown that transformers can be trained to perform simple regression tasks in-context. This work explores the possibility of training an in-context learner for classification tasks involving spurious features. We find that the conventional approach of training in-context learners is susceptible to spurious features. Moreover, when the meta-training dataset includes instances of only one task, the conventional approach leads to task memorization and fails to produce a model that leverages context for predictions. Based on these observations, we propose a novel technique to train such a learner for a given classification task. Remarkably, this in-context learner matches and sometimes outperforms strong methods like ERM and GroupDRO. However, unlike these algorithms, it does not generalize well to other tasks. We show that it is possible to obtain an in-context learner that generalizes to unseen tasks by training on a diverse dataset of synthetic in-context learning instances.

## 1 Introduction

Large language models, such as GPT-3, have the ability of in-context learning (ICL), wherein they learn to solve a task given a few examples in the context (Brown et al., 2020). The most significant aspect of in-context learning is that the learning happens during the forward pass on the context and query, without updating network parameters. In order to study in-context learning in isolation, a number of studies considered training transformers (Vaswani et al., 2017) from scratch to solve simple learning tasks in-context. In particular, Garg et al. (2022) show empirically that transformers can be trained to perform in-context learning of simple regression functions, such as dense or sparse linear functions, two-layer ReLU neural networks, and small decision trees.

Training on ICL instances can be seen as an instance of meta-learning (Schmidhuber, 1987; Naik and Mammone, 1992; Thrun and Pratt, 1998), where the goal is to learn a learning algorithm. What exact algorithm is learned when training transformers on ICL instances is still an open problem. Akyürek et al. (2022) and Von Oswald et al. (2023) show that transformers can implement a single gradient descent step of ordinary least squares and update the closed-form solution of ridge regression when a new example is added. Additionally, they provide evidence that transformers trained on ICL instances of linear regression learn algorithms that closely match predictions of the known algorithms, such as gradient descent on ordinary least squares objective and ridge regression. However, there is evidence that the learned algorithm may vary with model scale, depth, and pretraining task diversity (Akyürek et al., 2022; Raventós et al., 2024). In particular, Raventós et al. (2024) demonstrate that in the setting of in-context learning of linear regression tasks with insufficient pretraining task diversity, the learned algorithm behaves like a Bayesian estimator with the pretraining task distribution as the prior, and hence fails to generalize well to unseen tasks. Yadlowsky et al. (2023) show that when trained on ICL instances where the regression function belongs to a union of distinct function classes, the learned algorithm fails to generalize beyond the pretraining function classes. Ahuja and Lopez-Paz (2023) show that in-context learning ability diminishes under strong distribution shifts.

In this work, we explore the limits of in-context learning further by testing it on challenging settings. We deviate from the existing literature and consider visual classification tasks instead of regression tasks with simple function classes. In particular, we consider classification tasks where some features are *spuriously correlated* with the label. Such features are predictive of the label but are not causally

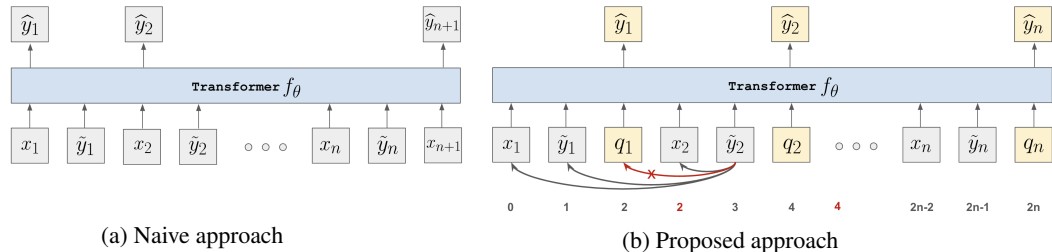

(a) Naive approach  (b) Proposed approach

Figure 1: In-context learning transformer architectures of the naive and proposed approaches. The proposed approach allows arbitrary query tokens after each learning example. Token positions and attention mask are modified so that these intermediate queries have no effect on other tokens.

related to it, due to which their correlation might not hold at test time. A prominent example is the cow vs camel classification task, where the background often correlates with the label, as cows are typically photographed in pastures, while camels are typically photographed in deserts (Beery et al., 2018). It is well-known that neural networks trained with gradient-based methods to minimize empirical risk can exploit spurious features, causing performance degradation under distribution shifts affecting these correlations (Torralba and Efros, 2011; Ribeiro et al., 2016; Gururangan et al., 2018; Zech et al., 2018; McCoy et al., 2019; Geirhos et al., 2019; 2020; Xiao et al., 2021).

We start our analysis in the standard setting of having a single classification task with spurious features. We consider the conventional approach of obtaining an in-context learner, wherein a transformer is trained on sequences of form $(x_1, y_1, \ldots, x_k, y_k, x_{k+1})$ to predict the label $y_{k+1}$ of the query example $x_{k+1}$. We find that this conventional approach leads to models that do classification *ignoring the context*, essentially memorizing the task. Furthermore, these models lack robustness to changes of the correlation between the label and spurious features. In particular, we observe a significant performance drop when the query follows a distribution in which the label and spurious feature correlation is zero. We propose an effective approach of addressing the task memorization issue. Namely, we find that task memorization can be mitigated greatly by randomly permuting input embedding dimensions for each training sequence. To address the issue of spurious features, we propose a novel way of forming ICL instances and a suitable transformer architecture, which work together to simulate distribution shift with respect to spurious features in the context. Overall, our proposed techniques lead to strong in-context learners that outperform established methods such as 1-NN, empirical risk minimization (ERM), and GroupDRO (Sagawa* et al., 2020), suggesting that the in-context learner implements a more specialized algorithm.

Despite being trained on instance of a single task, the learned algorithm generalizes to other tasks *without* spurious features. However, it fails to generalize to unseen tasks with spurious features. For this reason, we next explore training an in-context learner that generalizes to unseen tasks with spurious features. We create a dataset of in-context learning instances for various binary classifications tasks with varying spurious features. We demonstrate the efficacy of the proposed techniques on this dataset too and find that it can be improved further by passing spurious feature annotations as input and injecting occasional queries requesting the label of a proceeding context example to promote learning induction heads. The resulting model generalizes perfectly to unseen tasks, as long as the data generating process is similar. However, generalization to unseen tasks with possibly different data generating process depends on the severity of the challenge posed by spurious features, indicating that the learned algorithm is more brittle to severe distribution shifts than conventional algorithms. The source code for reproducing our experiments is available at `anonymized`.

We summarize our main contributions as follows.

(i) We show that the conventional approach of training an in-context learner is susceptible to presence of spurious features and also leads to task memorization in case of a single task.

(ii) We propose a suite of novel techniques of forming in-context training data to mitigate task memorization and increase robustness to spurious features, leading to in-context learners that outperform established learning algorithms.

(iii) We demonstrate that it is possible to obtain more general-purpose robust in-context learners by training on a diverse set of synthetic classification tasks involving spurious features.

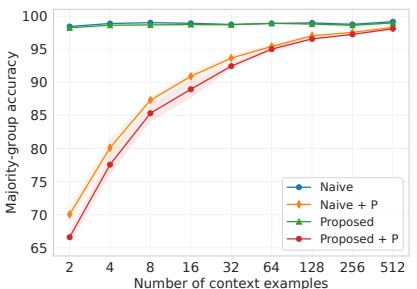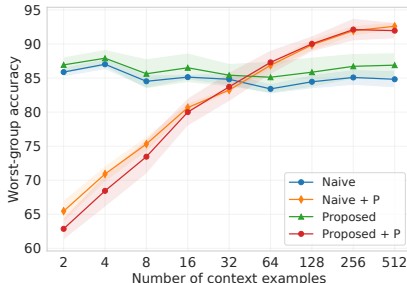

Figure 2: Majority-group and worst-group test accuracies on `Waterbirds` as a function of context size for the naive and proposed approaches with or without permuting input dimensions. Shaded regions show standard deviation across 5 training runs.

## 2 IN-CONTEXT LEARNING BASED ON A SINGLE TASK

We start by considering the common setting of having a single classification task with spurious features. For simplicity, we focus on label-balanced binary classification tasks in presence of a single binary spurious feature, although what follows next applies to label-imbalanced multiclass settings as well. Let $\mathcal{D}_{\text{train}}$ be a set of training examples for the task, where each example is a triplet $(x, s, y)$ of input $x \in \mathbb{R}^d$, spurious feature value $s \in \{0, 1\}$, and label $y \in \{0, 1\}$. Similarly, let $\mathcal{D}_{\text{test}}$ be a set of test examples. Importantly, we do not make any assumptions on the data generating process, except that $x$ has some information about $s$ and $s$ is predictive of $y$ on the training set, but their correlation does not hold on the test set. For an example $(x, s, y)$, we define its *group* $g = 2y + s$. In a binary classification task with a single binary spurious feature, there are four groups. Without loss of generality, we assume that for a majority of training examples we have that $y = s$. Hence we refer to groups 0 and 3 as majority groups, while referring to groups 1 and 2 as minority groups.

Training a transformer to perform linear regression in-context requires millions of ICL training instances, even for small dimensional cases. For example, Garg et al. (2022) use 32 million training instances for 20-dimensional inputs. We next consider ways of generating so many ICL instances from a single task.

### 2.1 A NAIVE APPROACH OF CONSTRUCTING ICL INSTANCES

The standard approach of constructing an ICL instance is to sample a subset of $n + 1$ examples $\{(x_i, s_i, y_i)\}_{i=1}^{n+1}$ from $\mathcal{D}_{\text{train}}$ and form a sequence $S = (x_1, \tilde{y}_1, x_2, \tilde{y}_2, \ldots, x_n, \tilde{y}_n, x_{n+1})$, where $\tilde{y}_i \in \mathbb{R}^d$ is a fixed random representation of either $y_i$ or $g_i$ (this distinction will be elaborated later). Then one trains a transformer $f_\theta : \cup_k \mathbb{R}^{k \times d} \to [0, 1]$ to predict $y_i$ given $S_i \triangleq (x_1, \tilde{y}_1, \ldots, x_{i-1}, \tilde{y}_{i-1}, x_i)$ (see Figure 1a), optimizing the following loss function:

$$\frac{1}{n+1} \sum_{i=1}^{n+1} \text{CE}(y_i, f_\theta(S_i)), \tag{1}$$

where $\text{CE}(y, \widehat{y}) = -y \log \widehat{y} - (1 - y) \log(1 - \widehat{y})$ is the binary cross-entropy loss. We explore two options of setting $\tilde{y}_i$. In the first option, we set $\tilde{y}_i$ to represent $y_i$ with a constant vector or its negative in $\mathbb{R}^d$. In this case we aim to obtain an in-context learner that is robust to spurious features without receiving spurious feature annotations as input. ERM is one such learner that minimizes average loss on training examples and does not require spurious feature annotations. In the second option, we set $\tilde{y}_i$ to represent $g_i$ as a sum of two constant vectors in $\mathbb{R}^d$, one representing the class and the other representing the spurious feature. In this case we aim to obtain an in-context learner that does robust classification with respect to a specified spurious feature. GroupDRO is one such learner that minimizes worst-group loss, therefore requiring spurious feature annotations at training time.

Unfortunately, the simple approach of (1) has several issues. First, as the classification task is the same in all ICL instances, the model can ignore context examples and predict $y_i$ based solely on $x_i$, essentially memorizing the task. Second, as all $n + 1$ examples of a sequence $S$ are sampled

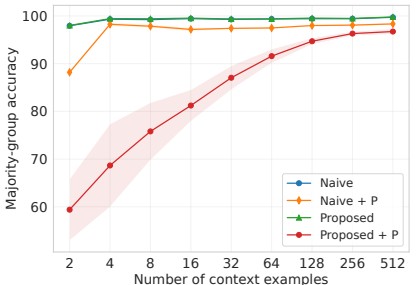 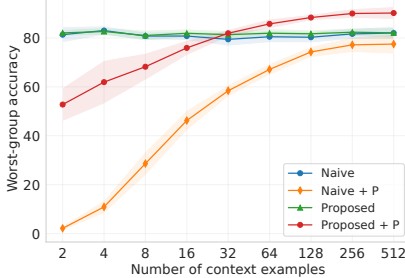

Figure 3: Majority-group and worst-group test accuracies on `Waterbirds-severe` as a function of context size for the naive and proposed approaches with or without permuting input dimensions.

from the training set and the spurious correlation holds for all of them, there is nothing preventing usage of spurious features in making predictions. To confirm these two issues, we consider the `Waterbirds` dataset (Sagawa* et al., 2020), which is landbird vs waterbird image classification task where image background (sea or land) is correlated with the label in the training set (4,795 examples), but not in validation and test sets . A robust classifier should predict `waterbird` or `landbird` without relying on image background. To separate out the representation learning challenge, we represent images with a pretrained and frozen DINOv2 ViT-B/14 distilled (Oquab et al., 2023). This way each image is embedded in $\mathbb{R}^{768}$. While using powerful pretrained representations increases overall performance under distribution shifts (Radford et al., 2021; Mehta et al., 2022), we note that it does not eliminate the problem of spurious correlations. Representations obtained via large-scale self-supervised pretraining are likely rich enough to capture information about both the label and spurious feature. Furthermore, many works have indicated that the main contribution to the out-of-domain generalization error comes from the classification head (rather than the representation learning module) and called for designing better methods of training the classification head (Galstyan et al., 2022; Menon et al., 2021; Kirichenko et al., 2023; Izmailov et al., 2022; Shi et al., 2023).

We train a causal decoder-only GPT-J transformer (Wang and Komatsuzaki, 2021) with 80M parameters on 2M in-context learning sequence with $n = 512$ and $\tilde{y}_i$ representing labels, constructed from the training set of `Waterbirds`. We use balanced sampling of classes and set the minority group proportion to 10% within each class. We use the ADAM optimizer (Kingma and Ba, 2014) ($\beta_1 = 0.9$ and $\beta_2 = 0.999$) with 32 batch size and no weight decay. The learning rate is selected from $\left\{3 \cdot 10^{-5}, 6 \cdot 10^{-5}, 10^{-4}\right\}$ based on average test performance over 5 runs. Concretely, we evaluate on 8192 sequences where the context part is $n$ training examples, while the query is a sampled from the test set with equal group distribution. Exact metric definitions and missing details are provided in Appendix A. Note that with 512 context length and 10% minority group ratio within each class, the expected value of the number of context examples from each of the 2 minority groups is about 25. For reference, the smallest minority-group has only 56 examples in the `Waterbirds` training set.

Figure 2 plots majority-group and worst-group test accuracies as a function of context size $n$. We see that naive approach results in models that ignore context – worst-group accuracy with 512 context examples is essentially the same as with 2 examples (see the *naive* curve). This confirms the task memorization issue. Figure 2 also shows that majority-group test accuracy of the naive approach is considerably higher compared to worst-group accuracy confirming the non-robustness issue.

## 2.2 THE PROPOSED APPROACH OF CONSTRUCTING ICL INSTANCES

To address the task memorization issue, we propose to rotate image embeddings in each ICL instance independently, making it harder to memorize individual examples. We found that generating random rotation matrices on fly is computationally expensive and slows down training. We tried generating and storing 10K rotation matrices, but this resulted in less than 50M different training examples that were still possible to memorize to some extent. A more effective and efficient alternative is to apply random permutations to image embedding dimensions (for brevity, this technique is denoted with +*P* in figures and tables; please see Figure 11 for an illustration of this technique). We found this approach to be very effective in terms of inducing in-context learning (see *naive + P* in Figure 2). We

also see that the difference between majority-group and worst-group accuracies decreases, although an approximately 5 p.p. gap remains.

When training an ICL transformer, ideally, we would like to simulate the situation of making a test prediction based on a context of training examples. Importantly, we would like to simulate the case where test distribution has balanced groups (i.e., the spurious correlation does not hold). Given access to spurious feature annotations *for the training set*, we can simulate this scenario using only training examples. In particular, we can form ICL instances of form $(x_1, \tilde{y}_1, \ldots, x_n, \tilde{y}_n, x_{n+1})$, where the context examples $(x_1, \ldots, x_n)$ are sampled in a way that the spurious feature is correlated with the label, while the query $x_{n+1}$ is sampled to have a uniform group distribution. However, if we again optimize the loss of (1), for context lengths less than $n$, the network will be allowed to make predictions using the spurious feature, which is undesirable. Please refer to Figure 17 of Appendix B for evidence of this. Potential ways of addressing this issue is upweighting the final prediction loss in Eq. (1) or upweighting predictions on minority examples. In our preliminary experiments we found the former approach ineffective. We did not experiment with the latter approach.

Instead, we propose a novel way of forming in-context learning instances and a modified transformer architecture that is suitable for such sequences. In particular, we form sequences of form $S = (x_1, \tilde{y}_1, q_1, x_2, \tilde{y}_2, q_2, \ldots, x_n, \tilde{y}_n, q_n)$, where $(x_i, \tilde{y}_i)$ are context examples, while $q_i$ are queries, sampled with replacement from $\mathcal{D}_{\text{train}} \setminus \{x_1, \ldots, x_n\}$. Importantly, $q_i$ are sampled with a uniform group distribution. Redefining $S_i = (x_1, \tilde{y}_1, q_1, \ldots, x_i, \tilde{y}_i, q_i)$, we would like the prediction on $S_i$ to be the label of $q_i$. When making a prediction on $q_i$, we want $q_j$ $(j < i)$ to have no effect. For this end we make two modifications. First, we modify the causal attention matrix to disallow attending to query tokens, unless a query token is attending to itself. Formally, if we enumerate tokens from 1 to $3n$ and define $M_{i,j}$ to denote the attention mask for token $i$ attending to token $j$, then we set

$$M_{i,j} = \begin{cases} 0, & i < j, \\ 0, & i > j \text{ and } j \equiv 0 \mod 3, \\ 1, & \text{otherwise.} \end{cases} \tag{2}$$

Second, we use modified token positions for computing positional encodings, in order to discount intermediate query tokens. Namely, for the sequence $(x_1, \tilde{y}_1, q_1, x_2, \tilde{y}_2, \ldots, x_n, \tilde{y}_n, q_n)$, position indices are set to $(0, 1, 2, 2, 3, 4, 4, \ldots, 2n - 2, 2n - 1, 2n)$. Formally, enumerating tokens from 1 to $3n$, the position index of the $i$-th token is set to $2 \lfloor \frac{i-1}{3} \rfloor + (i - 1) \mod 3$. Please refer to Figure 1 for an illustration. Hereafter, we refer to this approach as simply "proposed approach".

Figure 2 compares the proposed and naive approaches with and without input dimension permutations. Without random permutations, the proposed approach outperforms the naive approach marginally. However, the same is not true with random permutations. We found that image embeddings of DINOv2 have a bias towards representing objects more than backgrounds, alleviating the challenge posed by the spuriously correlated background in `Waterbirds`. In fact, the linear probing accuracy of the spurious feature is just $\approx 82\%$. For this reason, we create a modified version of Waterbirds by adding a constant vector $\tilde{s}$ or $-\tilde{s}$ to image embeddings based on the spurious feature $s$. We scale $\tilde{s}$ to have its norm equal to the average norm of image embeddings and verify that the linear probing accuracy of the spurious feature becomes 100%. On this modified `Waterbirds` dataset, which we name `Waterbirds-severe`, we see a large separation between the naive and proposed approaches (see Figure 3). We also see that without permutations, both naive and proposed approaches perform identically, indicating no robustness to the spurious correlation. This is expected, because in the absence of in-context learning, we can think of the naive and proposed approaches as standard and reweighted empirical risk minimization with a complex classification head, respectively. It has been observed that sample reweighting is not effective in overparameterized settings as all training examples will be perfectly fitted (Byrd and Lipton, 2019; Menon et al., 2021).

## 2.3 COMPARISON WITH CONVENTIONAL LEARNING ALGORITHMS

Now that we have established the efficacy of the proposed technique, we compare it to a few established algorithms, such as 1-NN, ERM, and GroupDRO, that last of two have been historically hard to outperform (Gulrajani and Lopez-Paz, 2021; Koh et al., 2021). Comparing to more existing methods designed for robustness to spurious correlations is outside of the goal of this work, namely studying limits of in-context learning. In our comparisons, we follow the evaluation recipe used for

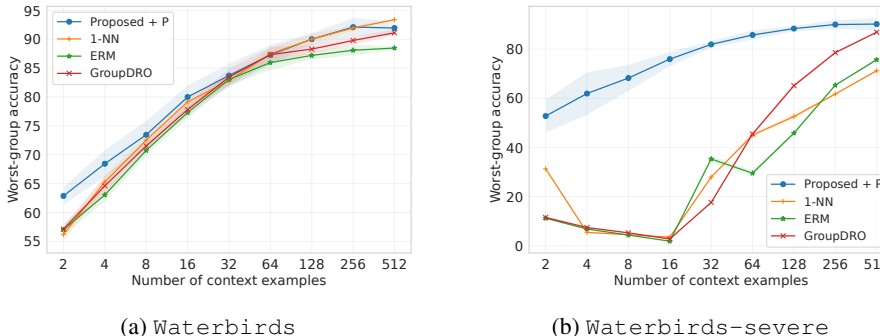

(a) Waterbirds

(b) Waterbirds-severe

Figure 4: Worst-group test accuracies on Waterbirds and Waterbirds-severe for the proposed approach and conventional methods such as 1-NN, ERM, and GroupDRO. Majority-group accuracies are reported in Figure 18 of Appendix B.

the in-context learners. Namely, we evaluate each baseline on 8192 sequences by training on the context part of the sequence and making a prediction on the single query. More information about hyperparameters and model selection is presented in Appendix A.

Figures 4a and 4b compare the proposed and baseline approaches on Waterbirds and Waterbirds-severe respectively. On Waterbirds, the proposed method outperforms ERM and GroupDRO on almost all context lengths, but is better than 1-NN only for short context lengths. The good performance of 1-NN is due to the bias in DINOv2 representations. On Waterbirds-severe, the proposed method outperforms the baselines at all context lengths. From these results, we conclude that this in-context learner implements none of these algorithms.

It is worth noting that baseline worst-group accuracies at $n = 512$ are actually *higher* than what we get when training on the entire dataset. For example, on Waterbirds, 1-NN gets only 90.03 % worst-group accuracy, while ERM gets 84.23 ± 0.17 % and GroupDRO gets 92.43 ± 0.24 %. This is due to balanced sampling of classes and setting the minority ratio to 10% withing each class, which is higher than the minority ratio of ≈ 5% in the original Waterbirds dataset. One can think of our resampling as a weaker form of down-sampling which has been found to be helpful in presence of spurious correlations (Nagarajan et al., 2021; Menon et al., 2021; Idrissi et al., 2022).

Additionally, we verify our findings on another popular dataset CelebA (Liu et al., 2015) designed for blond vs non-blond person classification, with sex being a spurious variable. Unlike, Waterbirds, the spurious feature is asymmetric in CelebA, as blond and non-blond women are equally represented, while blond men are significantly infrequent compared to non-blond men. In particular, we verify the two shortcomings of the conventional approach and demonstrate the efficacy of the proposed techniques compared to the baselines (please see Table 4 and Figure 16 of Appendix B).

## 2.4 GENERALITY OF THE LEARNED ALGORITHM

Since we train in-context learners on ICL instances of a single task, a natural question arises whether the learned algorithm can generalize to unseen tasks. Without permuting input dimensions, the model does not learn to do in-context learning. Thus, we can not hope for any generality without permuting input dimensions. We take the model obtained with the "Proposed + P" technique and probe generality of its in-context learning by evaluating on various datasets. We start by swapping the labels of two classes in Waterbirds at evaluation and observe ≈ 2 p.p. overall accuracy drop and ≈ 5 p.p. worst-group accuracy drop. Despite the worsened performance, this indicates that the model treats class labels symbolically, which is remarkable given that the semantics of labels were constant during training. However, when we evaluate on Waterbirds-severe, it gets 100% accuracy on the majority groups and *0% accuracy on minority groups*. Additionally, when we switch the task to predicting the background in the original Waterbirds dataset (now the class becomes a spurious feature), the overall test accuracy drops to 54.4%, while the worst-group accuracy drops to 9.3%.

It is worth noting that the learned algorithm is not completely useless for other tasks and works well in absence of spurious features, even on unseen tasks. For example, evaluating on binary classification

tasks derived from the CUB-200 (Welinder et al., 2010) dataset, from where the bird images of Waterbirds were taken, we get 99.7% accuracy at context size 100 (the accuracy is so high because most pairs of classes are easy to distinguish). We also test on binary classification tasks derived from classes belonging to *Amphibia* and *Mammalia* supercategories of the iNaturalist (Van Horn et al., 2018) dataset. At context length 512, the overall accuracy is 98.5%.

These OOD evaluation results indicate that the learned algorithm does something specific to the spurious feature of Waterbirds. We hypothesize that it learns to ignore this particular spurious feature. To test this, we evaluate on *group-balanced* Waterbirds sequences, with the task set to predicting background, and get 58.5% overall accuracy and 41.3% worst-group accuracy. Additionally, we do a forward pass on 1024 ICL Waterbirds sequences and collect final query representations at various layers of the transformer. We then do a linear probing (512 examples for probe training and 512 for validation) to measure predictability of the background variable. We find that the "Proposed + P" approach reduces background information effectively as we sweep from input to the final layer, while the "Naive" fails to reduce background probing accuracy (see Figure 21).

One potential way of improving generality and possibly also performance, is passing example groups as input, i.e., setting $\tilde{y}_i$ to represent $g_i$. We did not observe performance improvements and increase of generality of the learned algorithm when passing groups as input (see the complete results in Tables 1 and 2 of Appendix B). Thus, we conclude that when all ICL instances are derived from the task, the learned algorithm is inherently tied to the spurious feature of that task.

## 3 IN-CONTEXT LEARNING BASED ON A DIVERSE SET OF TASKS

In Section 2, we showed that it is possible to obtain a good in-context learner for a given task, but it fails to generalize to tasks with different spurious features. A better in-context learner should detect spurious features from context and make predictions without employing them. In this section, we explore the possibility of obtaining such a learner by training on a diverse set of ICL tasks. Since there exist few suitable datasets, we synthesize binary classification tasks with a single binary spurious feature, aiming to capture "structure" present in existing datasets. In short, given a standard binary classification task, say cat vs dog classification, for a sampled minority of cats we overwrite some of their features with those of random dogs. Similarly, we do an analogous operation for a sampled minority of dogs. This way some cats share dog features and vice versa. To create a diverse pool of in-context learning instances, we vary the two classes and the subset of grafted features. Please refer to Figure 15 of Appendix A for an illustration of this grafting operation.

More concretely, we consider the iNaturalist dataset (Van Horn et al., 2018), which contains images from 5,089 natural fine-grained categories and filter out categories that have less than 500 images. For testing purposes, from remaining 239 categories we set apart categories that belong to the supercategories *Amphibia* and *Mammalia*, along with 10% of random categories. We denote the set of these 48 categories as $\mathcal{C}_{ood}$, and the set of remaining 191 categories as $\mathcal{C}_{id}$, which we use to create in-context learning instances for training. For each category in $\mathcal{C}_{id}$, we hold out half of the examples as in-distribution validation set. To generate a single in-context learning instance, we sample two distinct classes from $\mathcal{C}_{id}$ randomly and sample $n/2$ images from the training split of each class uniformly at random without replacement. Please refer to Figure 14 of Appendix A for an illustration of our preprocessing of iNaturalist. We then do the grafting operation, setting minority group ratio within each class to 10%. We select the grafted features randomly, by first picking subset size $k$ uniformly at random from 0 to 199, and then sampling a random subset of embedding dimensions of size $k$. With this we get $n$ examples that form the context part of the instance. Abandoning the naive approach and focusing on the proposed one, for each class we sample $n/2$ queries from the remaining examples uniformly at random with replacement and do the grafting operation with 50% minority group ratio.

Following the experiments in Section 2, we train the same transformer with the proposed approach on 4M ICL instances with $n = 400$ context examples. We use the same optimizer and sweep the learning rate in the same range, selecting the best value based on the average *minority-group accuracy* (defined exactly in Appendix B) on instances where both categories belong to $\mathcal{C}_{ood}$ and thus were not observed during training. The results presented in Figure 5 indicate a major difference compared to the results in the single-task regime – namely, the proposed approach learns to do in-context learning to some extent without permuting embedding dimensions. As expected, we see much better performance with

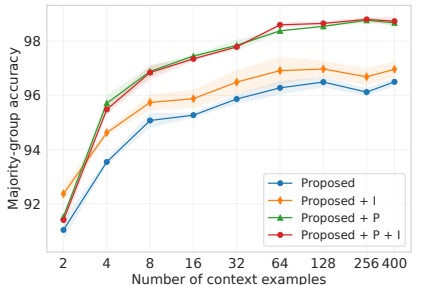 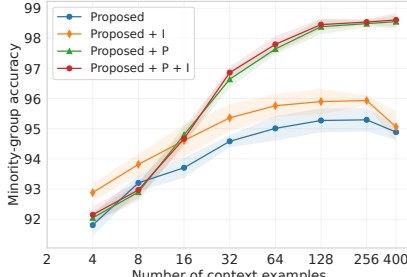

Figure 5: Majority-group and minority-group accuracies on the OOD test set of `iNaturalist` for the proposed approaches with or without permuting input dimensions and promoting induction heads.

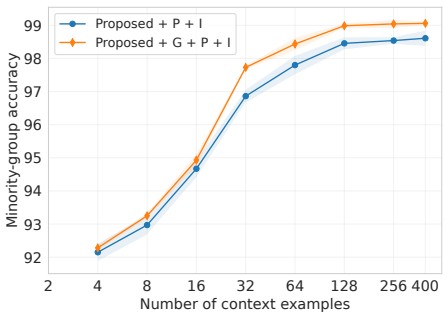 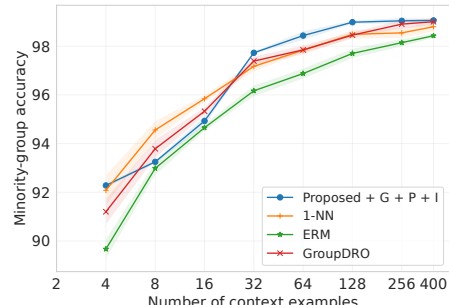

Figure 6: Minority-group accuracy on the OOD test set of `iNaturalist` for the best proposed approach with or without passing group information as input.

Figure 7: Minority-group accuracy on the OOD test set of `iNaturalist` for the best variant of proposed approach and conventional methods such as 1-NN, ERM, and GroupDRO.

permuted embedding dimensions. Notably, comparing majority-group and minority-group accuracies of the proposed approach with permutations, we see almost no sign of reliance on spurious features.

**Promoting emergence of induction heads.** In-context learning ability has been linked to induction heads, which are specific type of circuits found within large language models that implement the operation of looking back over the sequence for finding previous instances of the current token and copying what comes after that (Olsson et al., 2022). Inspired by this, we propose a data preparation technique that promotes learning of induction heads. With probability $p$, we replace each intermediate query independently with a random example from the proceeding part of the context (please see Figure 13 for an illustration of this technique). Note that this type of "hinting" is not possible in the naive approach and is enabled by the introduction of intermediate queries. In all experiments with this technique enabled, we just set $p = 0.25$. We observed that training of typical runs escapes the initial loss plateau faster with this technique (in about 3k iterations compared instead of about 10k iterations). Moreover, we see modest performance gains in `iNaturalist` experiments (see Figure 5, where +*I* stands for this technique).

**Passing example groups as input.** In contrast to the findings in the single-task setting of Section 2, we observed that setting $\tilde{y}_i$ to represent group improves the proposed approach, even on top of permitting input dimensions and promoting induction heads. One case of this is presented in Figure 6, while more cases can be found in the complete results presented in Appendix B. For brevity, we mark passing groups as inputs with +*G* in figures and tables. Please see Figure 12 for an illustration.

**Comparison with conventional learning algorithms.** Similar to the experiments in Section 2, we compare the best variant of the proposed approach (G + P + I) to 1-NN, ERM, and GroupDRO. Results presented in Figure 7 show that the learned algorithm is on-par with or outperforms the baselines starting at context length 32. The results at context lengths below 20 are not as informative, because the way we implemented the grafting operation implies that no examples are grafted when there are less than 10 examples in a class.

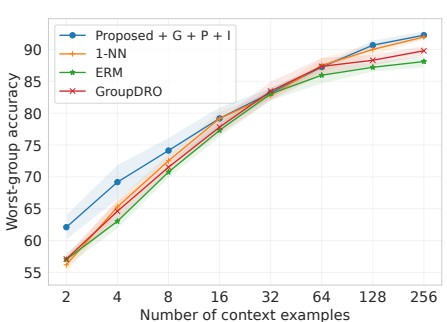

Figure 8: Worst-group test accuracy on `Waterbirds` for the best variant of proposed approach trained on `iNaturalist` and for methods such as 1-NN, ERM, and GroupDRO.

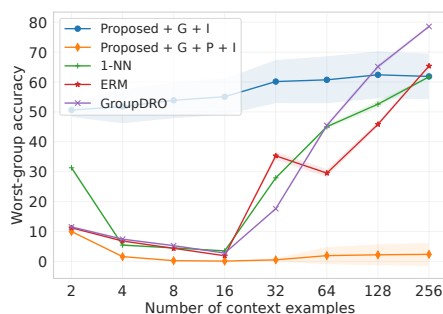

Figure 9: Worst-group test accuracy on `Waterbirds-severe` for the best variant of proposed approach trained on `iNaturalist` and for methods such as 1-NN, ERM, and Group-DRO.

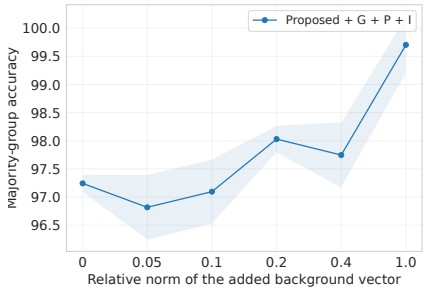
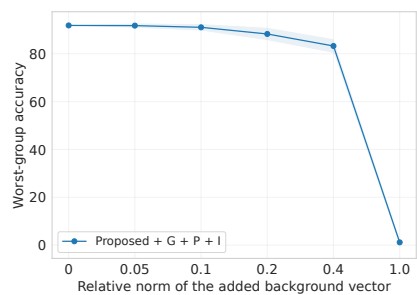

Figure 10: Majority-group and worst-group test accuracies of a proposed model (G + P + I) trained on `iNaturalist`, but evaluated on a modified variants of `Waterbirds` where we add a vector representing the spurious feature (background). The x-axis is the relative norm of the added vector compared to the average `Waterbirds` image embedding norm. Relative norm of 0 corresponds to `Waterbirds`, while relative norm of 1 corresponds to `Waterbirds-severe`.

**Generality of the learned algorithm.** To test the generality of the learned algorithm, we report evaluation results on `Waterbirds` (Figure 8) and `Waterbirds-severe` (Figure 9). We see that the learned algorithm outperforms baselines on `Waterbirds` and is as good as we got by training on `Waterbirds` itself. However, the learned algorithm fails completely on `Waterbirds-severe`, while the baselines give meaningful results starting at context length 32. We hypothesize that the challenge posed by the spurious features in `Waterbirds-severe` is significantly more severe compared to that in `iNaturalist`. By varying the norm of the added background vector, we interpolate between `Waterbirds` and `Waterbirds-severe`, and we see good generalization until the norm of the added vector is ≈40% of the average embedding norm (see Figure 10).

## 4 RELATED WORK

In this section, we discuss more related work in addition to the ones discussed earlier.

**In-weights vs in-context learning.** We observe two modes of learner behavior in our experiments. In the first mode, the learner acts like a standard supervised classifier, ignoring context examples. This mode appears when training on ICL instance of a single task without permuting input embedding dimensions. In the second mode, the learner does proper in-context learning. Our experiments indicate that both permuting embedding dimensions and increasing the number of training tasks are reliable ways of steering the model towards the in-context learning mode. The former is akin to the method of randomly projecting inputs proposed by Kirsch et al. (2022) for obtaining general-purpose in-context classifiers. Prior work has made a distinction between these two models of learning, naming them in-weights and in-context learning. In particular, Chan et al. (2022) demonstrate that certain

distributional properties of data, such as long-tail of class frequencies and bursty distribution of context example classes, can promote in-context learning when meta-training on few-shot classification instances. Singh et al. (2024) show that in-context learning behavior is not persistent and decays away with overtraining, indicating a trade-off between in-weights and in-context learning mechanisms. Moreover, they find that this in-context learning skill decay can be prevented by applying weight decay of embeddings and MLP layers, slowing down in-weight learning. Anand et al. (2024) make similar observations about these two modes of learning and propose active forgetting of token embeddings as an effective way of steering towards the in-context learning mode.

**Many shot ICL.** One ancillary finding of this work is that transformers can be trained to do in-context learning of visual classification tasks when good image embeddings are provided. This is remarkable because the input dimensionality we considered is much higher than what was considered in the pioneering works of Garg et al. (2022) and Akyürek et al. (2022) (784 vs 20). Furthermore, we observe predictable performance gains from longer context sizes. The number of "shots" we consider (up to 512 examples) is well beyond what is typically considered in ICL works (up to a few dozen of examples). Our findings are complementary to those of Agarwal et al. (2024), Jiang et al. (2024), and Li et al. (2024) who find that multimodal large language models, such as Gemini-1.5 Pro and GPT-4o, can benefit from large number of in-context demonstrations (up to 1000 demonstrations).

**In-context learning for out-of-distribution generalization.** Closest to our work are the works that propose to make use of in-context learning for out-of-distribution generalization. Han et al. (2023) test multimodal large language models (MLLMs) on a variety of visual classification tasks. They propose to leverage in-context learning abilities of MLLMs to improve performance on specialized domains and on tasks with significant corruptions. However, they only consider the case where both context examples and query are from the target domain. Zhang et al. (2024) make similar observations, but additionally study robustness of in-context learning to distribution shifts, such as domain shifts, label shifts, and spurious correlations. They find that in-context learning is highly susceptible to label shifts and presence of spurious correlations. Finally, Gupta et al. (2024) propose to address the problem of domain generalization (Muandet et al., 2013) by training an in-context learner that can take examples from a domain/environment and adapt to that domain in-context.

## 5  DISCUSSION AND CONCLUSION

We showed that it is possible to train an effective in-context learner tailored to a particular classification task with spurious features. We did this by introduce two key techniques: (a) permuting input embedding dimensions and (b) forming ICL sequences with intermediate queries simulating distribution shift. We provided evidence that the learned algorithm is highly competitive on the task it was trained on. However, we found that while it generalizes to other tasks without spurious features, it does not work for tasks with other spurious features. Understanding this failure mechanistically and exploring techniques for enabling better generalization are key future research directions.

We next explored training on synthetic ICL instances of diverse tasks and showed that it is possible to obtain an in-context learner that generalizes to unseen tasks, even with different data generating processes. We established the usefulness of two more techniques: (c) passing example groups as input and (d) promoting learning of induction heads by occasionally querying past context examples. We believe there is a room for improving in-context learning via improved strategies of choosing intermediate queries and possibly optimizing worst-group loss. Understanding why the learned algorithm fails under extreme distribution shifts and why variants with permutations fail more (see Figure 10) is an interesting question to explore. Another interesting direction to explore is to find out what exact algorithm is learned in the process of training on diverse tasks. Based on the results presented in this work, we conclude that the learned algorithm is neither 1-NN, ERM, or GroupDRO.

Our work has several limitations. First of all, training a transformer-based in-context learner with high-dimensional image embeddings is computationally costly (see Appendix A for information on compute resources), although it is faster than the baselines at inference. For this reason, we did not explore more datasets and pretrained image embeddings. We believe main conclusions of our work will be unchanged and provide an experiment on `CelebA` with a larger network in Appendix B. Second, we experimented with only one model size, width, and depth. Larger models might behave differently (Wei et al., 2023). Third, in our `iNaturalist` experiments, we considered only one

"type" of spurious features. It is likely that this choice has significant effect on the learned algorithm and its generality. Future research should explore more ways of synthesizing spurious features and consider varying severity of the challenge posed by spurious features. The latter can be done by considering multiple spurious features, introducing label imbalance, varying magnitude of spurious correlations, and varying the margin spurious features provide.

Finally, we acknowledge that the proposed approach of *training* robust in-context learners requires spurious feature annotations, which is typically costly to obtain. As we have shown, this limitation can be addressed by creating synthetic data, in which case spurious annotations are readily available. At inference, the learned algorithm does not require spurious annotations if it is trained with $\tilde{y}_i$ set to represent $y_i$ (i.e., ERM-like algorithm), but requires when it is trained with passing example groups as input (i.e., $\tilde{y}$ set to represent $g_i$; GroupDRO-like algorithm). It is important to note that as we consider classification problems where the learner is given training data only from *environment*, spurious annotations are, in general, necessary to disambiguate core and spurious features.

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

# A FURTHER EXPERIMENTAL DETAILS

**Baselines.** For empirical risk minimization as a baseline, we tune 2 hyperparameters: learning rate (0.01 or 0.001) and number of epochs (100 or 200). For GroupDRO we additionally tune its parameter that controls adaptiveness of group weights (0.01, 0.1, or 1) and we also try an optional strong L2 regularization (1.0 weight decay), as it has been observed to be useful for small datasets (Sagawa* et al., 2020).

**Transformer-based methods.** In all transformer-based approaches, we train a causal decoder-only GPT-J transformer with 80M parameters that has 6 transformer layers with 8 multi-head attention, 768 model dimensionality, and 3072 hidden dimensionality. When training on `iNaturalist`, we add a layer normalization (Ba et al., 2016) on transformer input, as we expect input norms to change when we evaluate on `Waterbirds`-based datasets. The transformer input sequence in the proposed approach consists of 3 types of tokens: context image embeddings, query image embeddings, and label/group annotations. While the network can rely on positions and content to distinguish image embeddings from annotations, we found it to be helpful to encode token types explicitly. We do this by setting the first 3 dimensions of a token to be a one-hot vector representing token type (context image embedding, query image embedding, or annotation). When permuting dimensions, we do the permutation before encoding token types to keep the location of token-type information consistent. In our preliminary experiments and development, we used $n = 128$ context length. Apart from improved performance, we did not observe significant qualitative differences when we switched to larger context lengths for final experiments.

**Evaluation and model selection.** For all transformer-based approaches and baselines, we do a grid search to find the best combination of hyperparameters. In particular, we train each configuration with 5 different random seeds and selected one with the highest average test performance. Importantly, for baseline methods model selection is done for each context length independently, while for transformer-based methods model selection is done once with respect to the test performance at maximum context length observed during training. All evaluations are done on 8192 sequences, where the first $n$ examples are sampled from the corresponding train set while the query is sampled from the test set with a balanced group distribution. Finally, even when training transformers on permuted image embeddings, we do not apply permutations during evaluation. In all figures throughout this work, shaded regions show standard deviation across the 5 training runs.

Note that the most principled model selection approach would be selecting models based on a metric calculated on a dataset similar to the training set (e.g., a held-out part of training set), rather than the test set. For example, in the case of experiments on `Waterbirds` or `Waterbirds-severe`, the principled approach would be to select based on performance on sequences where the context part is sampled from the training set, while the final query is sampled from a held-out validation set with balanced group distribution. We tried this way of model selection and did not observe significant changes. In the case of experiments on `iNaturalist`, the principled approach would be to select based on performance on sequences where the context part is sampled from the training set, while the final query is sampled from the hold-out part the training set. We observed that this in-distribution metric is always around 99.5%-100%, and can be non-informative for model selection. This is a typical scenario in OOD generalization (see for example (Gulrajani and Lopez-Paz, 2021) or (Wenzel et al., 2022)).

**Definitions of metrics.** Given a set of predictions on `Waterbirds` or `Waterbirds-severe`, worst-group accuracy is defined as the lowest accuracy of predictions among the 4 groups. Note that worst-group accuracy is not applicable to `iNaturalist`, as different ICL sequences correspond to different classification tasks and hence form different groups. For this reason, we introduce minority-group and majority-group accuracies. Given a triplet $(C, q, \widehat{y})$, where $C$ is a context, $q$ is query, and $\widehat{y}$ is a prediction on $q$, we call $\widehat{y}$ a minority (majority) prediction, if $q$ is among the least (most) represented group(s) of the context $C$. Given a list of triplets $(C, q, \widehat{y})$, we define minority (majority) group accuracy as the accuracy among minority (majority) predictions.

**Compute resources.** We used NVIDIA A100 GPUs with 40GB memory to train transformer-based methods. The network we considered is small enough to fit on one GPU with batch size 32 when $n = 400$ (`iNaturalist` experiments) and batch size 24 when $n = 512$ (`Waterbirds`

and `Waterbirds-severe` experiments). We did mixed 16-bit training to save compute and did not notice any quality degradation. A single training takes around 12 hours for `iNaturalist` experiments and around 18 hours for `Waterbirds` experiments. We used a mix of CPUs and weaker GPUs to train baselines, as they are not computationally as demanding.

ICL instance #1: embedding dimension permutation = (2, 3, 1, 4, 6, 5)

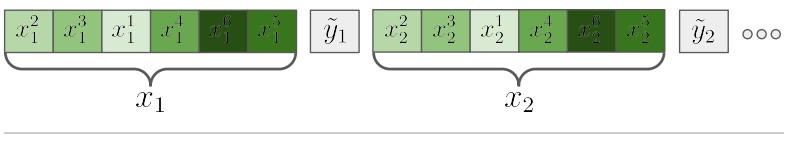

ICL instance #2: embedding dimension permutation = (1, 6, 5, 3, 4, 2)

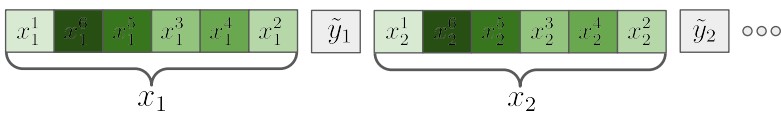

Figure 11: An illustration of the proposed technique of permuting image embedding dimensions (denoted with *+P* throughout the paper). Note that in each ICL instance we sample a new permutation, but the same permutation is used to permute dimensions of all image embeddings within one ICL instance. Whenever, we use the proposed approach of forming ICL sequences (see Figure 1b), the dimensions of intermediate queries are also permuted.

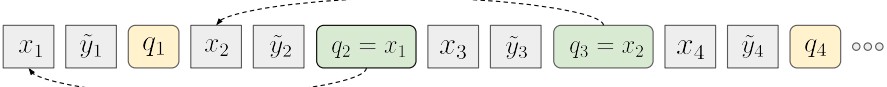

Figure 12: An illustration of the proposed approach with passing example groups as input (denoted with *+G* throughout the paper).

$$x_1 \quad \underbrace{(2y_1 - 1)u + (2s_1 - 1)v}_{\tilde{y}_1 = \tilde{g}_1} \quad q_1 \quad x_2 \quad \underbrace{(2y_2 - 1)u + (2s_2 - 1)v}_{\tilde{y}_2 = \tilde{g}_2} \quad q_2 \quad \circ\circ\circ$$

$$x_1 \quad \tilde{y}_1 \quad q_1 \quad x_2 \quad \tilde{y}_2 \quad q_2 = x_1 \quad x_3 \quad \tilde{y}_3 \quad q_3 = x_2 \quad x_4 \quad \tilde{y}_4 \quad q_4 \quad \circ\circ\circ$$

Figure 13: An illustration of the proposed approach with promoting emergence of induction heads (denoted with *+I* throughout the paper). Intermediate queries that are randomly selected to be one of the previous context examples are shown in green.

# B  ADDITIONAL RESULTS

In addition to the figures presented in the main text, here we provide the exact experimental resources for multiple transformer-based and baseline approaches, some of which were not included in the main text due to space constraints. Recall that +P means permuting input dimensions, +I means promoting learning of induction heads, and +G means passing example groups as input to in-context learning transformers.

Table 1 presents worst-group accuracies on the test set of `Waterbirds` for 3 sets of approaches: (a) in-context learners trained on `Waterbirds` itself, (b) in-context learners trained on `iNaturalist`, and (c) baselines. Similarly, Table 2 presents worst-group accuracies on the test set of `Waterbirds-severe` for 3 sets of approaches: (a) in-context learners trained on `Waterbirds-severe` itself, (b) in-context learners trained on `iNaturalist`, and (c) baselines. As RoPE-based transformers are not good at length extrapolation (Press et al., 2021), we do not attempt evaluating models trained on `iNaturalist` with context size 400 on 512-long sequences of `Waterbirds` or `Waterbirds-severe`. Finally, Table 3 presents minority-group accuracy

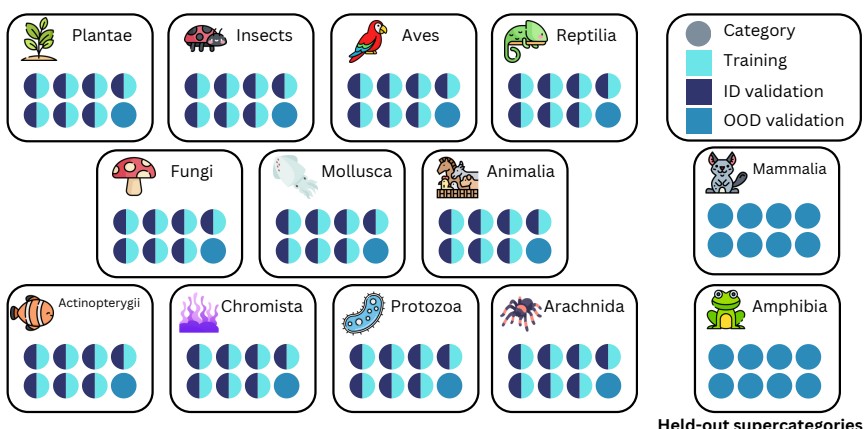

Figure 14: An illustration of our preprocessing of the iNaturalist dataset.

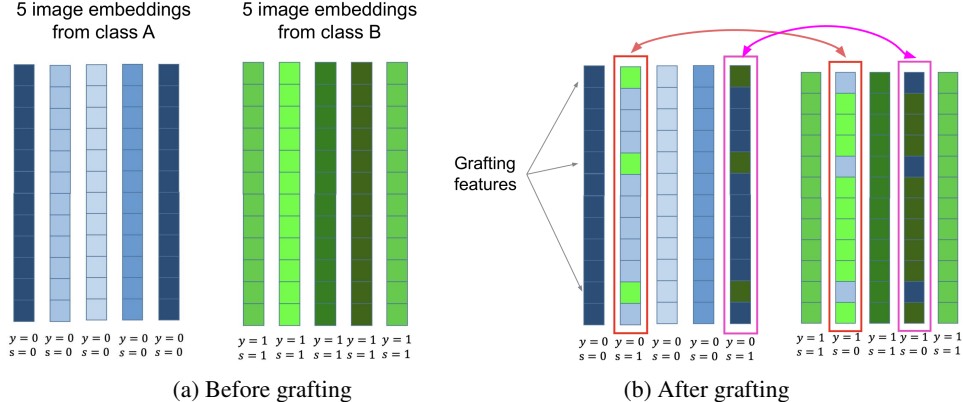

(a) Before grafting        (b) After grafting

Figure 15: An illustration of the grafting operation for creating spurious features. The figure (a) depicts two classes of examples, each having 5 examples given by 12-dimensional embeddings. In this example, the grafting operation selects 3 embedding dimensions to become spurious features. For this end, these 3 features of examples 2 and 5 of class A are swapped with those of examples 2 and 4 of class B, respectively. Figure (b) depicts the embeddings after the grafting operation.

on out-of-distribution classes of iNaturalist for two sets of approaches: (a) in-context learners trained on iNaturalist itself and (b) baselines.

**Experiments on CelebA.** To further verify our main findings presented in Section 2, we conduct experiments on another popular visual classification tasks CelebA (Liu et al., 2015). In CelebA, the task is to classify blond vs non-blond persons, with sex being a spuriously correlated variable. Notably, the spurious correlation is asymmetric, in the sense that blond and non-blond women are almost equally represented, while blond men are much less represented compared to non-blond men. We follow the design of Waterbirds experiments in our CelebA experiments, with the only difference that we set the group distribution of context examples to $(0.25, 0.25, 0.05, 0.45)$, where group 0 are non-blond men, group 1 are non-blond women, group 2 are blond men, and group 4 are blond women. Table 4 presents worst-group accuracies on the test set of CelebA for 2 sets of approaches: (a) in-context learners trained on CelebA itself and (b) baselines algorithms. As in our Waterbirds experiments, we see that it is essential to permute input embeddings *and* to form ICL sequences in the proposed fashion. Unlike Waterbirds, comparing "Proposed + P" with "Proposed + P + G" we see that providing spurious annotations in-context provides significant gains. Figure 16 demonstrates that both of these approaches outperform 1-NN, ERM, and GroupDRO.

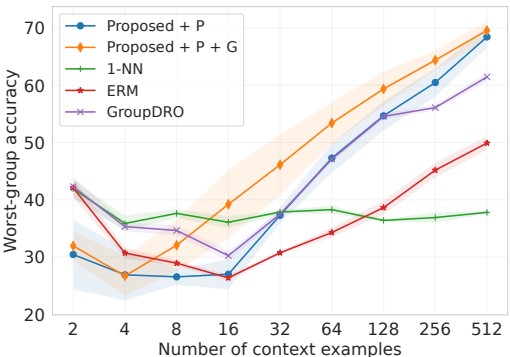

Figure 16: Worst-group test accuracies on `CelebA` for the proposed approach and conventional methods such as 1-NN, ERM, and GroupDRO. Shaded regions show standard deviation across 5 training runs.

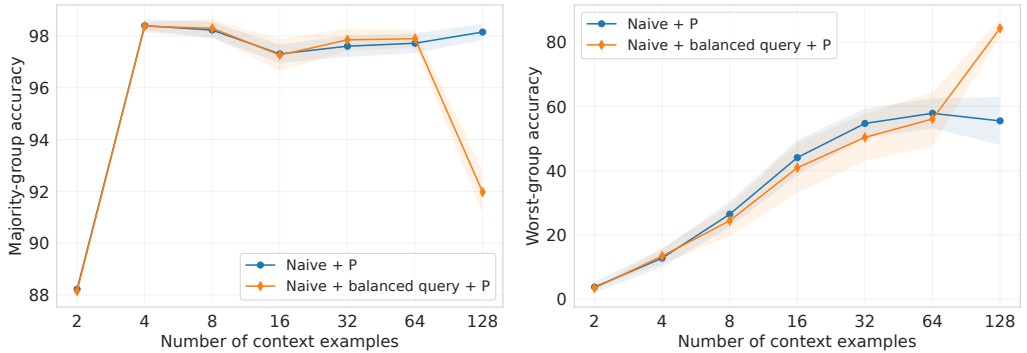

Figure 17: Majority-group and worst-group test accuracies on `Waterbirds-severe` as a function of context size for the naive approach with a single modification of making the last example (query) group-balanced. Shaded regions show standard deviation across 5 training runs. As expected, at intermediate context lengths this method performs similar to the naive approach, but is much better at the training context length.

**Experiments with a larger network.** To verify that our findings generalize to larger models, we repeat CelebA experiments but with a transformer architecture of 12 layers with 12 multi-head attention (instead of 6 layers with 8 multi-head attention). Due to memory increase, we decrease the batch size from 24 to 8. Besides these two changes, we keep all other experimental details the same. The complete results presented in Table 5 are qualitatively the same compared to the smaller network case (Table 4), with the difference that the results of transformer-based entries are lower. Furthermore, the standard deviation of the $+P$ approaches is significantly higher, indicating difficulties in optimization. We hypothesize that this is due to reusing learning rate and training length that were that were tuned for the smaller network with 3 times larger batch size.

**On data leakage in single task regime.** In the single task setting of Section 2, there is a potential for data leakage, not in the sense that individual examples might be leaked (we always evaluate on unseen examples), but in the sense that the learner effectively observes more data from the single task than its context length at evaluation. Indeed, when we do not permute input embeddings, we observe task memorization (i.e., data leakage) and the model does very well at evaluation with even close to empty context. To verify that there is no data leakage when we enable permuting input embeddings (+P), we take one of the "Proposed + P" runs trained on `Waterbirds` and evaluate it on ICL sequences where input embeddings of each sequence are rotated with a random *rotation matrix*. As the set of permutation matrices is a measure-zero subset of general rotation matrices, we expect that in case of data leakage we would observe degraded performance, as the model would be

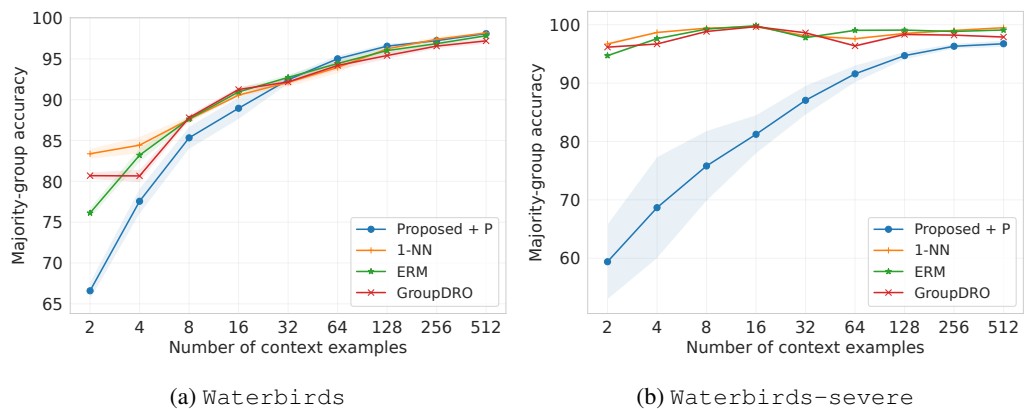

(a) Waterbirds          (b) Waterbirds-severe

Figure 18: Majority-group test accuracies on Waterbirds and Waterbirds-severe for the proposed approach and conventional methods such as 1-NN, ERM, and GroupDRO. Shaded regions show standard deviation across 5 training runs.

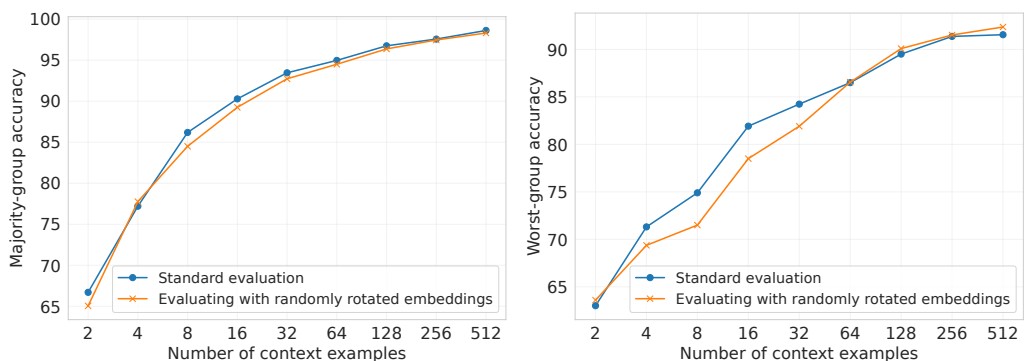

Figure 19: Majority-group and worst-group test accuracies on Waterbirds as a function of context size for a "Proposed + P" run evaluated on ICL sequences with randomly rotated input embeddings. Largely unchanged evaluation results fail to confirm that there is any data leakage when input embeddings are permuted during training.

expecting randomly permuted embeddings of some memorized embedding space. In results presented in Figure 19, we see that under this new evaluation the results are the same (up to statistical noise), failing to confirm that there is any data leakage when input embeddings are permuted during training. Finally, note that data leakage is not a concern in the multiple task setting of Section 3, because we evaluate on either unseen categories of iNaturalist or on unseen tasks such as Waterbirds and Waterbirds-severe.

**Experiments with group-balanced contexts.** As noted in Section 5, the proposed approach of *training* an in-context learner requires spurious annotations. Given access to spurious annotations, one can simply train an in-context learner on sequences with balanced groups. While in-context learners obtained this way will not be useful for new tasks for which we do not have spurious annotations (and thus cannot form group-balanced contexts), it is still useful to compare how well this approach does in the single task setting of Section 2. For this end, we train in-context learners on balanced-group sequences consisting of 128 Waterbirds examples. This way each group is represented with 32 context examples. Note that in our main Waterbirds experiments with 512 context examples but group-imbalanced contexts, the minority groups are represented with even less, 25 examples. As the group-balanced sampling context breaks the correlation between the label and spurious feature, we only consider the naive approach of forming ICL sequences (Figure 2). The results presented in Figure 20 show that, as expected, group-balanced sampling improves worst-group accuracy. The naive approach, which again ignores the context and does tasks memorization, reaches $86.08 \pm$

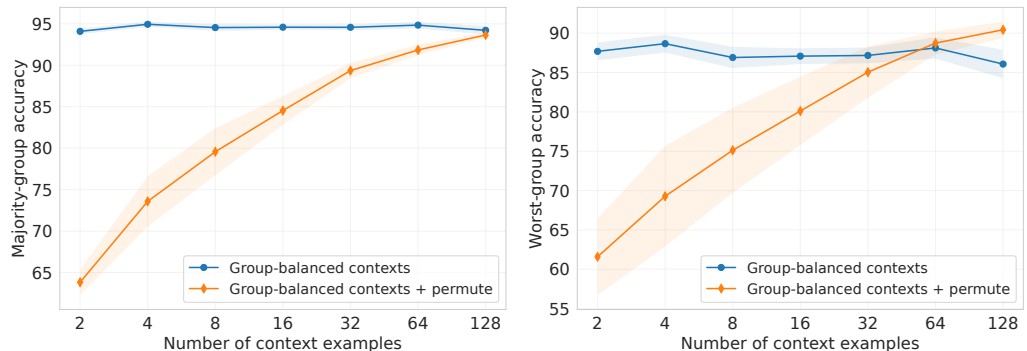

Figure 20: Majority-group and worst-group test accuracies on `Waterbirds` as a function of context size for the naive approach trained and evaluated on *group-balanced* contexts. The training is done on ICL sequences with 128 context examples. Shaded regions show standard deviation across 5 training runs.

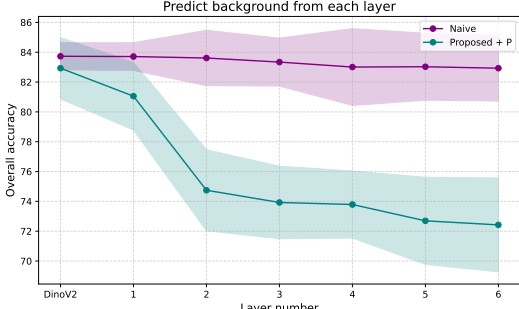

Figure 21: Linear probing accuracy of the background variable at various layers of in-context learner transformers trained on `Waterbirds`.

1.87 worst-group accuracy with 128 group-balanced context examples, compared to $84.82 \pm 1.26$ worst-group accuracy on 512 *group-imbalanced* context examples (see Table 1). This positive effect of downsampling has been also observed in standard (not in-context) training settings (Nagarajan et al., 2021; Menon et al., 2021; Idrissi et al., 2022). Furthermore, we again see that the proposed technique of permuting embedding dimensions induces strong in-context learning and reaches $90.44 \pm 1.10$ worst-group accuracy with 128 group-balanced context examples.

Table 1: Complete results on `Waterbirds`. Reported numbers are average worst-group test accuracies, along with the their standard deviation. The top half of in-context learners were trained on `Waterbirds` itself, while the ones in the bottom half were training on `iNaturalist`.

| Method / Context size | 4 | 8 | 16 | 32 | 64 | 128 | 256 | 512 |
|---|---|---|---|---|---|---|---|---|
| Naive | 87.02 (0.79) | 84.52 (1.00) | 85.14 (0.42) | 84.82 (0.89) | 83.41 (0.75) | 84.45 (1.04) | 85.08 (1.15) | 84.82 (1.26) |
| Naive + P | 70.92 (1.18) | 75.32 (1.11) | 80.66 (0.68) | 83.24 (0.35) | 86.87 (0.62) | 89.87 (0.85) | 91.94 (0.75) | 92.60 (0.59) |
| Proposed | 87.91 (1.29) | 85.63 (2.20) | 86.51 (2.17) | 85.42 (1.73) | 85.12 (2.34) | 85.86 (2.22) | 86.72 (1.89) | 86.89 (1.82) |
| Proposed + I | 88.18 (1.07) | 85.89 (1.31) | 86.68 (1.02) | 86.01 (1.39) | 84.82 (1.02) | 85.92 (1.23) | 86.07 (1.27) | 86.46 (1.57) |
| Proposed + P | 68.44 (2.40) | 73.46 (2.53) | 80.00 (2.06) | 83.71 (2.15) | 87.30 (1.79) | 90.02 (1.16) | 92.11 (1.65) | 91.95 (1.20) |
| Proposed + P + I | 68.05 (1.51) | 72.47 (1.80) | 78.97 (1.12) | 82.58 (0.68) | 86.39 (0.69) | 90.00 (0.60) | 91.78 (0.56) | 92.17 (0.86) |
| Proposed + G | 88.74 (1.01) | 87.00 (1.60) | 87.62 (1.58) | 86.86 (1.31) | 86.18 (1.33) | 86.91 (0.98) | 87.26 (1.11) | 86.95 (1.21) |
| Proposed + G + I | 88.89 (0.53) | 87.49 (0.69) | 87.70 (0.74) | 86.90 (0.95) | 86.03 (0.71) | 86.64 (0.72) | 87.29 (0.77) | 87.35 (1.00) |
| Proposed + G + P | 68.47 (2.32) | 73.74 (2.00) | 79.21 (1.68) | 82.85 (1.33) | 86.55 (1.17) | 89.98 (0.72) | 92.00 (0.82) | 93.05 (0.40) |
| Proposed + G + P + I | 68.24 (1.88) | 73.78 (1.67) | 80.23 (0.94) | 83.02 (1.22) | 86.94 (1.31) | 89.89 (0.91) | 92.46 (1.00) | 92.69 (1.15) |
| 1-NN | 65.29 (1.23) | 72.53 (1.11) | 79.15 (1.16) | 82.81 (0.63) | 87.49 (1.18) | 90.00 (1.05) | 91.96 (0.51) | 93.40 (0.27) |
| ERM | 63.04 (1.22) | 70.76 (1.01) | 77.32 (1.16) | 83.04 (1.09) | 85.95 (1.38) | 87.20 (0.77) | 88.10 (0.98) | 88.48 (0.45) |
| GroupDRO | 64.61 (1.79) | 71.52 (0.73) | 77.81 (1.19) | 83.45 (1.57) | 87.34 (1.42) | 88.30 (0.91) | 89.79 (0.81) | 91.12 (0.62) |
| Naive | 69.77 (1.37) | 77.98 (1.51) | 79.23 (0.83) | 81.20 (1.35) | 82.57 (1.52) | 83.85 (1.56) | 84.21 (1.19) | - |
| Naive + P | 66.47 (1.17) | 73.12 (1.44) | 77.85 (1.74) | 81.76 (1.49) | 86.36 (0.86) | 88.02 (1.25) | 89.68 (0.77) | - |
| Proposed | 69.75 (5.51) | 77.51 (3.01) | 79.20 (2.11) | 81.39 (1.49) | 82.04 (1.29) | 83.51 (0.97) | 84.63 (0.80) | - |
| Proposed + I | 70.73 (1.42) | 77.10 (1.76) | 78.90 (1.49) | 80.86 (1.74) | 82.22 (1.72) | 84.22 (1.45) | 84.69 (1.47) | - |
| Proposed + P | 66.09 (1.49) | 73.71 (1.17) | 78.33 (0.69) | 82.75 (0.83) | 86.32 (0.52) | 88.85 (0.72) | 89.98 (1.35) | - |
| Proposed + P + I | 65.51 (2.16) | 70.91 (2.32) | 75.94 (3.04) | 81.51 (1.90) | 86.41 (1.50) | 89.39 (0.98) | 91.08 (0.75) | - |
| Proposed + G | 70.98 (2.52) | 78.41 (1.25) | 79.67 (1.26) | 81.59 (1.42) | 82.42 (1.28) | 83.91 (1.64) | 84.31 (1.31) | - |
| Proposed + G + I | 71.94 (2.70) | 78.56 (1.65) | 80.62 (1.66) | 82.31 (1.76) | 83.52 (1.57) | 84.52 (1.32) | 85.35 (1.20) | - |
| Proposed + G + P | 67.55 (0.78) | 73.79 (0.33) | 78.32 (0.93) | 82.56 (1.31) | 86.01 (1.09) | 89.40 (1.22) | 90.99 (1.15) | - |
| Proposed + G + P + I | 69.18 (2.76) | 74.13 (2.06) | 79.18 (1.81) | 83.17 (0.85) | 87.25 (0.37) | 90.67 (0.80) | 92.23 (0.69) | - |

Table 2: Complete results on `Waterbirds-severe`. Reported numbers are average worst-group test accuracies, along with the their standard deviation. The top half of in-context learners were trained on `Waterbirds-severe` itself, while the ones in the bottom half were training on `iNaturalist`.

| Method / Context size | 4 | 8 | 16 | 32 | 64 | 128 | 256 | 512 |
|---|---|---|---|---|---|---|---|---|
| Naive | 83.04 | 80.78 | 80.78 | 79.43 | 80.50 | 80.29 | 81.67 | 82.02 |
|  | (1.92) | (1.58) | (1.85) | (2.77) | (2.43) | (2.30) | (2.25) | (2.72) |
| Naive + P | 10.89 | 28.61 | 46.23 | 58.40 | 67.13 | 74.28 | 77.18 | 77.49 |
|  | (2.71) | (4.98) | (4.17) | (2.46) | (2.34) | (2.25) | (3.11) | (4.08) |
| Proposed | 82.64 | 81.01 | 81.90 | 81.36 | 81.94 | 81.70 | 82.35 | 82.09 |
|  | (1.56) | (2.23) | (1.80) | (1.69) | (1.91) | (1.62) | (1.72) | (2.15) |
| Proposed + I | 83.23 | 80.76 | 81.65 | 81.46 | 81.63 | 81.34 | 81.46 | 82.24 |
|  | (1.30) | (1.93) | (2.38) | (2.11) | (2.40) | (2.01) | (2.32) | (3.49) |
| Proposed + P | 61.94 | 68.23 | 75.94 | 81.93 | 85.76 | 88.36 | 90.01 | 90.20 |
|  | (8.91) | (5.53) | (3.13) | (1.53) | (2.03) | (1.30) | (1.98) | (2.65) |
| Proposed + P + I | 64.01 | 72.22 | 78.45 | 82.00 | 85.86 | 88.13 | 90.09 | 90.59 |
|  | (4.05) | (4.43) | (2.79) | (2.20) | (1.64) | (1.39) | (1.73) | (1.54) |
| Proposed + G | 82.02 | 81.15 | 83.11 | 81.22 | 81.30 | 81.90 | 82.48 | 82.44 |
|  | (3.37) | (3.56) | (1.84) | (2.08) | (1.72) | (1.62) | (1.62) | (1.34) |
| Proposed + G + I | 82.61 | 80.48 | 81.20 | 80.13 | 81.09 | 80.84 | 81.61 | 81.84 |
|  | (3.42) | (2.69) | (3.55) | (3.18) | (2.86) | (2.47) | (2.36) | (2.51) |
| Proposed + G + P | 59.11 | 64.44 | 71.30 | 79.46 | 85.21 | 88.60 | 90.65 | 91.38 |
|  | (2.89) | (5.67) | (3.74) | (0.83) | (1.54) | (1.36) | (1.01) | (1.14) |
| Proposed + G + P + I | 64.26 | 70.05 | 77.76 | 82.38 | 86.56 | 89.09 | 90.75 | 90.82 |
|  | (5.81) | (4.01) | (1.77) | (1.66) | (0.88) | (1.02) | (0.96) | (0.73) |
| 1-NN | 5.44 | 4.50 | 3.49 | 27.92 | 45.04 | 52.58 | 61.74 | 71.20 |
|  | (0.60) | (0.43) | (0.21) | (0.54) | (0.88) | (1.39) | (0.48) | (0.58) |
| ERM | 6.81 | 4.35 | 1.87 | 35.30 | 29.52 | 45.84 | 65.35 | 75.69 |
|  | (0.44) | (0.26) | (0.24) | (1.55) | (1.49) | (1.00) | (0.53) | (0.88) |
| GroupDRO | 7.42 | 5.26 | 2.75 | 17.62 | 45.47 | 65.13 | 78.57 | 86.89 |
|  | (0.57) | (0.35) | (0.29) | (0.65) | (1.18) | (1.06) | (0.77) | (0.57) |
| Naive | 48.18 | 49.39 | 48.71 | 52.58 | 54.10 | 56.41 | 56.86 | - |
|  | (3.52) | (3.28) | (6.49) | (4.56) | (6.04) | (5.03) | (4.75) |  |
| Naive + P | 0.88 | 0.06 | 0.00 | 0.13 | 0.19 | 0.13 | 0.02 | - |
|  | (0.45) | (0.05) | (0.00) | (0.29) | (0.43) | (0.29) | (0.04) |  |
| Proposed | 49.04 | 53.39 | 54.82 | 59.44 | 61.04 | 62.26 | 63.77 | - |
|  | (2.76) | (4.74) | (8.82) | (10.75) | (12.23) | (12.31) | (12.38) |  |
| Proposed + I | 48.45 | 52.44 | 54.74 | 58.67 | 60.37 | 62.42 | 63.27 | - |
|  | (6.15) | (11.15) | (10.69) | (11.38) | (9.19) | (8.69) | (9.15) |  |
| Proposed + P | 1.88 | 0.27 | 0.06 | 0.08 | 0.30 | 0.15 | 0.03 | - |
|  | (0.56) | (0.21) | (0.10) | (0.13) | (0.60) | (0.28) | (0.06) |  |
| Proposed + P + I | 2.27 | 0.66 | 0.15 | 1.18 | 2.50 | 1.14 | 0.50 | - |
|  | (0.74) | (0.49) | (0.14) | (1.09) | (2.49) | (0.88) | (0.20) |  |
| Proposed + G | 50.00 | 52.31 | 53.69 | 57.87 | 59.11 | 60.33 | 62.30 | - |
|  | (5.03) | (5.05) | (4.54) | (3.33) | (3.16) | (3.36) | (3.01) |  |
| Proposed + G + I | 51.78 | 53.87 | 55.07 | 60.15 | 60.73 | 62.40 | 61.86 | - |
|  | (5.76) | (6.15) | (6.20) | (7.40) | (8.02) | (8.01) | (7.77) |  |
| Proposed + G + P | 1.52 | 0.16 | 0.00 | 0.10 | 0.04 | 0.03 | 0.36 | - |
|  | (0.69) | (0.13) | (0.00) | (0.20) | (0.05) | (0.05) | (0.73) |  |
| Proposed + G + P + I | 1.59 | 0.23 | 0.08 | 0.50 | 1.91 | 2.19 | 2.34 | - |
|  | (0.17) | (0.16) | (0.10) | (0.69) | (3.05) | (3.67) | (4.00) |  |

Table 3: Complete results on `iNaturalist`. Reported numbers are average minority-group accuracies on the OOD test set of `iNaturalist`, along with the their standard deviation.

| Method / Context size | 4 | 8 | 16 | 32 | 64 | 128 | 256 | 400 |
|---|---|---|---|---|---|---|---|---|
| Proposed | 91.80 (0.39) | 93.20 (0.29) | 93.71 (0.35) | 94.58 (0.22) | 95.01 (0.42) | 95.27 (0.42) | 95.30 (0.40) | 94.89 (0.27) |
| Proposed + I | 92.88 (0.31) | 93.82 (0.37) | 94.61 (0.56) | 95.36 (0.45) | 95.76 (0.40) | 95.90 (0.44) | 95.94 (0.18) | 95.06 (0.54) |
| Proposed + P | 92.04 (0.22) | 92.90 (0.30) | 94.80 (0.32) | 96.64 (0.30) | 97.65 (0.20) | 98.39 (0.27) | 98.49 (0.14) | 98.55 (0.23) |
| Proposed + P + I | 92.15 (0.28) | 92.97 (0.30) | 94.67 (0.28) | 96.86 (0.21) | 97.80 (0.29) | 98.46 (0.20) | 98.54 (0.11) | 98.61 (0.25) |
| Proposed + G | 92.48 (0.45) | 93.27 (0.72) | 93.88 (0.43) | 94.91 (0.63) | 94.99 (0.38) | 95.29 (0.45) | 95.13 (0.33) | 94.64 (0.43) |
| Proposed + G + I | 92.59 (0.33) | 93.80 (0.23) | 94.18 (0.38) | 95.50 (0.33) | 95.82 (0.41) | 95.83 (0.34) | 95.82 (0.55) | 95.28 (0.60) |
| Proposed + G + P | 91.90 (0.17) | 92.84 (0.19) | 94.69 (0.15) | 97.28 (0.31) | 98.29 (0.13) | 98.70 (0.19) | 98.85 (0.19) | 99.00 (0.11) |
| Proposed + G + P + I | 92.28 (0.10) | 93.25 (0.09) | 94.93 (0.22) | 97.73 (0.07) | 98.44 (0.20) | 98.99 (0.09) | 99.04 (0.14) | 99.06 (0.07) |
| 1-NN | 92.08 (0.64) | 94.56 (0.39) | 95.84 (0.16) | 97.17 (0.23) | 97.84 (0.12) | 98.49 (0.20) | 98.55 (0.23) | 98.80 (0.21) |
| ERM | 89.67 (0.43) | 92.98 (0.30) | 94.65 (0.17) | 96.17 (0.24) | 96.88 (0.23) | 97.70 (0.21) | 98.15 (0.17) | 98.43 (0.11) |
| GroupDRO | 91.20 (0.55) | 93.79 (0.39) | 95.33 (0.18) | 97.39 (0.20) | 97.85 (0.20) | 98.46 (0.13) | 98.91 (0.20) | 99.01 (0.18) |

Table 4: Complete results on `CelebA`. Reported numbers are average worst-group test accuracies, along with the their standard deviation. All in-context learning were train on `CelebA` itself.

| Method / Context size | 4 | 8 | 16 | 32 | 64 | 128 | 256 | 512 |
|---|---|---|---|---|---|---|---|---|
| Naive | 24.88 (2.03) | 24.56 (2.26) | 25.80 (1.98) | 25.14 (2.11) | 23.85 (1.91) | 25.62 (1.63) | 25.84 (2.10) | 26.20 (1.42) |
| Naive + P | 20.72 (2.21) | 17.27 (2.38) | 12.43 (2.04) | 14.85 (2.33) | 13.56 (1.60) | 16.17 (2.83) | 20.16 (4.14) | 26.03 (5.13) |
| Proposed | 25.83 (1.77) | 25.42 (1.66) | 26.85 (1.52) | 25.80 (1.60) | 25.18 (1.06) | 26.65 (2.11) | 26.89 (1.41) | 27.53 (1.31) |
| Proposed + P | 26.90 (4.56) | 26.54 (1.46) | 27.00 (2.72) | 37.30 (1.55) | 47.29 (2.67) | 54.66 (2.67) | 60.48 (2.55) | 68.45 (1.99) |
| Proposed + G | 23.87 (1.50) | 24.67 (1.51) | 25.60 (1.30) | 24.95 (1.12) | 24.42 (1.17) | 25.77 (1.24) | 25.70 (0.81) | 26.55 (1.39) |
| Proposed + G + P | 26.71 (3.57) | 32.06 (4.62) | 39.21 (6.06) | 46.13 (5.38) | 53.41 (3.73) | 59.37 (3.07) | 64.39 (1.84) | 69.58 (1.72) |
| 1-NN | 35.87 (1.48) | 37.63 (0.86) | 36.08 (1.13) | 37.86 (0.45) | 38.28 (0.80) | 36.40 (0.32) | 36.90 (0.99) | 37.81 (0.36) |
| ERM | 30.70 (1.05) | 28.93 (0.64) | 26.37 (0.66) | 30.75 (0.40) | 34.29 (0.93) | 38.64 (1.29) | 45.18 (1.39) | 49.92 (1.28) |
| GroupDRO | 35.32 (0.88) | 34.64 (0.99) | 30.24 (1.01) | 37.49 (0.60) | 47.11 (0.59) | 54.56 (0.66) | 56.11 (0.60) | 61.47 (0.93) |

Table 5: Complete results on `CelebA`, but with larger network of 120m parameters, consisting of 12 layers (instead of 6 layers) with 12 multi-head attention (instead of 8 heads). Reported numbers are average worst-group test accuracies, along with the their standard deviation. All in-context learning were train on `CelebA` itself.

| Method / Context size | 4 | 8 | 16 | 32 | 64 | 128 | 256 | 512 |
|---|---|---|---|---|---|---|---|---|
| Naive | 24.43 (0.57) | 21.79 (0.86) | 23.68 (0.58) | 23.02 (0.49) | 23.99 (0.84) | 23.18 (1.04) | 22.62 (0.83) | 20.22 (1.00) |
| Naive + P | 21.34 (1.40) | 14.64 (0.66) | 12.56 (0.74) | 13.35 (1.54) | 13.76 (2.58) | 15.73 (2.41) | 17.69 (3.27) | 21.67 (4.55) |
| Proposed | 22.90 (2.30) | 20.86 (2.40) | 23.11 (2.66) | 21.93 (2.50) | 23.35 (2.80) | 21.82 (3.11) | 21.75 (2.32) | 19.54 (2.06) |
| Proposed + P | 35.13 (5.19) | 31.54 (1.73) | 30.89 (4.40) | 35.19 (5.28) | 41.63 (6.74) | 47.81 (9.40) | 51.60 (11.00) | 55.53 (11.70) |
| Proposed + G | 23.08 (2.47) | 20.89 (2.66) | 22.74 (2.54) | 21.73 (3.23) | 22.70 (3.25) | 21.24 (3.45) | 21.50 (2.61) | 18.90 (2.01) |
| Proposed + G + P | 31.44 (4.27) | 32.09 (5.24) | 36.54 (1.90) | 43.67 (2.72) | 49.62 (2.93) | 54.77 (4.24) | 58.75 (5.94) | 61.74 (6.10) |
| 1-NN | 35.87 (1.48) | 37.63 (0.86) | 36.08 (1.13) | 37.86 (0.45) | 38.28 (0.80) | 36.40 (0.32) | 36.90 (0.99) | 37.81 (0.36) |
| ERM | 30.70 (1.05) | 28.93 (0.64) | 26.37 (0.66) | 30.75 (0.40) | 34.29 (0.93) | 38.64 (1.29) | 45.18 (1.39) | 49.92 (1.28) |
| GroupDRO | 35.32 (0.88) | 34.64 (0.99) | 30.24 (1.01) | 37.49 (0.60) | 47.11 (0.59) | 54.56 (0.66) | 56.11 (0.60) | 61.47 (0.93) |

