# OpenReview forum: "In-context learning in presence of spurious correlations"
_ICLR.cc/2025/Conference — Submitted to ICLR 2025_

### Official Review · Reviewer_WYsn · 2024-10-29

**Soundness:** 3
**Presentation:** 2
**Contribution:** 2
**Rating:** 5
**Confidence:** 4

**Summary:**

This paper studies in-context learning image classfication tasks in presence of spurious correlations. The authors find that:
* permuting the image embeddings prevents transformers from memorizing the task and promotes in-context learning;
* inserting intermediate queries along with permuting embeddings enhences robustness to spurious correlations;
* using proceeding examples as queries improves the performance on both majority and minority groups;
* passing group values to the context improves the performance on minority groups.

Except for Waterbirds dataset, these findings are also verified on (i) the Waterbirds-severe dataset where the spurious features are enhenced; (ii) a varying spurious correlations dataset adapted from iNaturalist. However, the transformer trained on dataset in (ii) fails to be robust against the spurious correlations in the dataset in (i).

**Strengths:**

This paper has a clear structure and interesting findings. I appreciate that authors also conduct experiments in the setting of different spurious correlations and test the trained model on unseen tasks.

**Weaknesses:**

1. Motivation for the proposed approach is unclear. I feel the effcacy of the proposed approach should be attributed to the fact that $q_i$ are sampled from the uniform group distribution and the autoregressive loss is only taken on these queries (Please correct me if I am wrong). This seems a bit werid to me -- under the spurious correlation setting, is it appropriate to assume that we have access to the uniform group distribution at the training stage? If one has access to the uniform group distribution data, why not simply use a training sequence consisting of these data as in-context examples?

2. Lacks of in-depth analysis and interpretation. For example, in section 2.4 the authors obtain interesting results by stwiching the evaluation data to (i) Waterbirds-severe; (ii) background prediction on Waterbirds; (iii) background prediction on Waterbirds with group-balanced context. There is one vague hypothesis saying the model learns to ignore the specific spurious feature in the pretrain data, which lacks further analysis and evidence. The trained model gets around 50% overall test accuracy and 9% worst-group accuracy on predicting the backgroud, which justifies your this hypothesis. However, the results on Waterbirds-severe and group-balanced Waterbirds seem to suggest the model still *in-context learns* to use the background feature. It feels to me the model learns some "biased initialization" (prefer to ignore specific feature) after pretraining on the spurious correlated data with the proposed approach, and then adjusts accroding to the in-context examples on top of this "biased initialization". Discussions along similar directions are lacked. Similar issues for the generality part in section 3. I feel more in-depth discussions would be beneficial.

3. Clarity issue: the paragraph "More concretely, ....." in section 3 is hard to parse. Would be helpful if there's an illustration for this data generating procedure. Also some other minor problems, please see questions.

**Questions:**

1. When producing Waterbirds-severe dataset, how is $\tilde s$ computed? Is it added before or after the permutation if using $+P$ technique.
2. For the $+I$ technique, is there any evidence that the performance gain is attributed to the forming of induction heads? This is not very obvious to me since the pair $(x_i,\tilde y_i)$ does not contribute to the loss function explicitly and the label of $q_{i-1}$ does not appear in the sequence.
3. In the last paragraph, the authors claims the proposed approach does not require spurious feature annotations at inference time. Am I missing anything? I feel using $\tilde y$ to represent group ($+G$) technique requires the annotations at inference time since the inference sequence is also using the same form of $\tilde y_i$.
4. The questions in the weakness part.

---

> ### Author Response · Authors · 2024-11-25
> **Author response (part 1)**
>
> We thank the reviewer for their careful review and thoughtful comments. We appreciate that the reviewer finds that our work has a clear structure and interesting findings. We provide responses to the reviewer’s specific questions below.
>
> > 1. Motivation for the proposed approach is unclear. I feel the efficacy of the proposed approach should be attributed to the fact that
>  are sampled from the uniform group distribution and the autoregressive loss is only taken on these queries (Please correct me if I am wrong). This seems a bit weird to me -- under the spurious correlation setting, is it appropriate to assume that we have access to the uniform group distribution at the training stage? If one has access to the uniform group distribution data, why not simply use a training sequence consisting of these data as in-context examples?
>
> Thank you for raising this subtle aspect. We have elaborated on this in the updated draft but address here too in detail. We agree that the efficacy of the proposed approach should be attributed to the fact that intermediate queries are sampled from a uniform group distribution, but would like to stress also the critical role of permuting image embeddings which induces strong in-context learning, avoiding task memorization.
>
> Regarding the question *"is it appropriate to assume that we have access to the uniform group distribution at the training stage?"*, it is important note that we are considering classification problems where the learner is given training data only from one environment/domain. Because the learner observes only a single environment, in general, there is no way for it to determine which features are spurious. This implies that in single domain settings, in general, one needs to specify the spurious feature, either implicitly through world knowledge and inductive biases or with explicit annotations.
>
> It is also helpful to distinguish the two cases: (a) training an in-context learner that does not take spurious annotations (i.e., $\tilde{y}$ represents $y$) and (b) training an in-context learner that does take spurious annotations as input (i.e., $\tilde{y}$ represents group
> $g$). As stated at lines 150-159, we can think of (a) as algorithms having the same signature as ERM and (b) as algorithms having the same signature as GroupDRO. At test time, when a learned ERM-like (type a) algorithm is applied to a new task, we do not need spurious feature annotations. However, when a learned GroupDro-like (type b) algorithm is applied to a new task, it requires spurious feature annotations, just as GroupDRO does.
>
> Regarding the question *"If one has access to the uniform group distribution data, why not simply use a training sequence consisting of these data as in-context examples?"*, indeed, in the single task setting (Section 2), one can train on contexts with balanced groups. However, ERM-like ($\tilde{y}$ representing $y$) learners trained on balanced contexts would be useless for related tasks with an unannotated spurious feature. This distinction becomes even more important in the multiple task setting (Section 3), where the goal is to learn an algorithm that is useful for new related tasks. Nevertheless, we agree that it is still useful to understand how well this approach does in the single task setting of Section 2. For this end, we trained in-context learners on balanced-group sequences consisting of 128 Waterbirds examples. This way each group is represented with 32 context examples. Note that in our main Waterbirds experiments with 512 context examples but group-imbalanced contexts, the minority groups are represented with even less, 25 examples. As the group-balanced sampling context breaks the correlation between the label and spurious feature, we only consider the naive approach of forming ICL sequences (Figure 2). The results presented in Figure 20 of the updated draft show that, as expected, group-balanced sampling improves worst-group accuracy. The naive approach, which again ignores the context and does tasks memorization, reaches 86.08 $\pm$ 1.87 worst-group accuracy with 128 group-balanced context examples, compared to 84.82 $\pm$ 1.26 worst-group accuracy on 512 group-imbalanced context examples (see Table 1)). This positive effect of downsampling has been also observed in standard (not in-context) training settings (Nagarajan et al., 2021; Menon et al., 2021; Idrissi et al., 2022). Furthermore, we again see that the proposed technique of permuting embedding dimensions induces strong in-context learning and reaches 90.44 $\pm$ 1.10 worst-group accuracy with 128 group-balanced context examples. This is just a bit below the 91.95 $\pm$ 1.20 worst-group accuracy we get training "Proposed + P" on 512 *group-imbalanced* context examples (see Table 1).

---

> ### Author Response · Authors · 2024-11-25
> **Author response (part 2)**
>
> > For example, in section 2.4 the authors obtain interesting results by switching the evaluation data to (i) Waterbirds-severe; (ii) background prediction on Waterbirds; (iii) background prediction on Waterbirds with group-balanced context. There is one vague hypothesis saying the model learns to ignore the specific spurious feature in the pretrain data, which lacks further analysis and evidence. The trained model gets around 50% overall test accuracy and 9% worst-group accuracy on predicting the background, which justifies your this hypothesis. However, the results on Waterbirds-severe and group-balanced Waterbirds seem to suggest the model still in-context learns to use the background feature. It feels to me the model learns some "biased initialization" (prefer to ignore specific feature) after pretraining on the spurious correlated data with the proposed approach, and then adjusts according to the in-context examples on top of this "biased initialization". Discussions along similar directions are lacked. Similar issues for the generality part in section 3. I feel more in-depth discussions would be beneficial.
>
> Thank you for raising this important aspect. We agree that more in-depth and mechanistic understanding of the learned algorithm would be helpful. Regarding the evaluation on Waterbirds-severe, we hypothesize that the failure is due to the newly added spurious vector, which the model has not learned to remove and uses for classification, resulting in a non-robust model. Regarding the evaluation on group-balanced Waterbirds for background prediction, the bad results indicate that indeed the model learned to ignore/remove background information. We have updated the draft and added an experiment confirming this hypothesis (see Figure 21). In particular, we did a forward pass on 1024 ICL \waterbirds{} sequences and collected final query representations at various layers of the transformer. We then did a linear probing (512 examples for training and 512 validation) to measure predictability of the background variable. We found that the "Proposed + P" approach reduces background information effectively as we sweep from input to the final layer, while the "Naive" fails to reduce background probing accuracy.
>
> > Clarity issue: the paragraph "More concretely, ....." in section 3 is hard to parse. Would be helpful if there's an illustration for this data generating procedure. Also some other minor problems, please see questions.
>
> We have updated the draft with an illustration of our preprocessing of the iNaturalist dataset (see Figure 14).
>
> > Question 1. When producing Waterbirds-severe dataset, how is s computed? Is it added before or after the permutation if using +P technique.
>
> We add the extra spurious feature ($\pm$ a constant vector) to all Waterbirds image embeddings before applying permutations.
>
> > Question 2. For the +I technique, is there any evidence that the performance gain is attributed to the forming of induction heads? This is not very obvious to me since the pair (x_i,\tilde{y}_i does not contribute to the loss function explicitly and the label of q_{i-1} does not appear in the sequence.
>
> Thank you for the good observation. We do not show it mechanistically, however there are two bits of evidence suggesting this. First, for a transformer model, the easiest way of learning queries that ask the label of a previous context example is implementing an induction head. Second, without promoting induction heads, we often observe a long loss plateau in the beginning of training. This long plateau goes away when we enable promoting induction heads (+I). Such plateaus have been connected to learning of induction heads (see for example Singh et al. (2024)).
>
> > Question 3. In the last paragraph, the authors claim the proposed approach does not require spurious feature annotations at inference time. Am I missing anything? I feel using y to represent group (+G) technique requires the annotations at inference time since the inference sequence is also using the same form of \tilde{y}_i.
>
> Thank you for spotting a mistake. We have corrected this in the updated draft. See the last paragraph of Section 5.

---

### Official Review · Reviewer_ZBp4 · 2024-11-03

**Soundness:** 2
**Presentation:** 3
**Contribution:** 2
**Rating:** 5
**Confidence:** 3

**Summary:**

The paper explores the challenges and limitations of training large language models for in-context learning (ICL) on classification tasks that involve spurious features. The authors argue that conventional ICL approaches are susceptible to spurious correlations and can lead to task memorization rather than leveraging context for predictions. To address these issues, the paper proposes novel techniques for training ICL models that are more robust to spurious features and less prone to task memorization. The proposed methods involve permuting input embedding dimensions and constructing ICL instances with intermediate queries to simulate distribution shifts. The paper demonstrates that these techniques can lead to in-context learners that match or outperform established algorithms like ERM and GroupDRO on certain tasks. Furthermore, the authors show that by training on a diverse dataset of synthetic ICL instances, it is possible to obtain a more general-purpose in-context learner that can generalize to unseen tasks involving spurious features.

**Strengths:**

1. The paper introduces a new perspective on ICL by focusing on the impact of spurious correlations, which is an important yet underexplored area in the context of large language models.
2. The authors provide a thorough analysis of the problem, backed by empirical evidence from experiments on various datasets, including Waterbirds, CelebA, and iNaturalist.
3. The proposed method to mitigate task memorization and increase robustness to spurious features are innovative and show promising results in improving ICL, especially when the number of context examples is limited.
4. The paper not only focuses on in-context learning for a specific task but also addresses the generalization of the learned algorithm to unseen tasks, which is a critical aspect of real-world applicability.
5. The paper is well-organized, with a clear presentation of the problem, methodology, experiments, and results, making it easy to follow the authors' line of reasoning.

**Weaknesses:**

My primary concern with this paper lies in the limited support for the effectiveness of the proposed methods: "(a) passing example groups as input and (d) promoting learning of induction heads by occasionally querying past context examples."  The paper lacks a strong theoretical foundation for these methods, and the experimental validation is somewhat restricted.

Specifically, I have the following observations:

1. The experiments consider only a single model structure, depth, and width. The impact of these architectural hyperparameters on the effectiveness of the proposed methods remains unexplored.  Investigating different architectures could reveal whether the observed benefits are consistent across a wider range of models.

2. The experimental results are confined to three datasets: Waterbirds, CelebA, and iNaturalist. While these datasets are commonly employed in the Out-of-Distribution (OOD) generalization literature, they are relatively simple and may not fully capture the complexity of spurious correlations encountered in real-world applications. Evaluating the proposed methods on more diverse and challenging datasets would strengthen the claim of their effectiveness.

3. The paper does not address the possibility of data leakage, which could inadvertently inflate the reported performance.  A thorough analysis of potential data leakage sources and mitigation strategies is necessary to ensure the validity of the results.

To enhance the paper, I recommend the authors consider the following:

1. Provide a theoretical foundation: Develop a theoretical framework that explains why the proposed methods are expected to mitigate spurious correlations. This could involve analyzing the underlying mechanisms through which these methods operate and providing formal guarantees or bounds on their performance.

2. Increase experimental diversity: Expand the experimental evaluation to include a wider range of model architectures, hyperparameters, and datasets. This would provide a more comprehensive assessment of the effectiveness and generalizability of the proposed methods.

3. Address potential data leakage: Thoroughly investigate and discuss potential sources of data leakage in the experimental setup. Implement appropriate mitigation strategies and report any remaining limitations.

By addressing these issues, the authors can provide a more compelling argument for the effectiveness of their proposed methods in addressing spurious correlations.

**Questions:**

Could the authors provide a visual analysis to help readers intuitively understand the specific spurious features on which the existing methods and the proposed methods are effective or ineffective?

---

> ### Author Response · Authors · 2024-11-24
>
> We thank the reviewer for their careful review and thoughtful comments. We appreciate that the reviewer finds our work well-written, addressing an important under-explored area, and presenting innovative techniques with promising results. We provide responses to the reviewer’s specific questions below.
>
> > 1. The experiments consider only a single model structure, depth, and width. The impact of these architectural hyperparameters on the effectiveness of the proposed methods remains unexplored. Investigating different architectures could reveal whether the observed benefits are consistent across a wider range of models.
>
> We agree it would be good to experiment with more architectures and model sizes, but this requires significant computational resources, as we do a full hyperparameter grid search with 5 independent runs for each hyperparameter setting. Nevertheless, we have conducted new experiments on Celeb-A with a larger network and observed qualitatively similar results. Please see Appendix B of the updated draft for this.
>
> > 2. The experimental results are confined to three datasets: Waterbirds, CelebA, and iNaturalist. While these datasets are commonly employed in the Out-of-Distribution (OOD) generalization literature, they are relatively simple and may not fully capture the complexity of spurious correlations encountered in real-world applications. Evaluating the proposed methods on more diverse and challenging datasets would strengthen the claim of their effectiveness.
>
> Again, conducting experiments on new datasets requires significant computation resources. Furthermore, as we consider binary classification problems with binary spurious features, there are not many real-world datasets that match this setting.
> Nevertheless, we are conducting experiments on a new dataset and will update if the results become available on time.
>
> > 3. The paper does not address the possibility of data leakage, which could inadvertently inflate the reported performance. A thorough analysis of potential data leakage sources and mitigation strategies is necessary to ensure the validity of the results.
>
> Thank you for raising a very important and a subtle concern. We have added clarifications and an experiment on this in the updated draft, but we will present them here too. We would like to state that there can be no data leakage by design in the multiple task setting of Section 3, because we evaluate on either unseen categories of iNaturalist or on unseen tasks such as Waterbirds and Waterbirds-Severe. In Section 2, there is a potential for data leakage, not in the sense that individual examples might be leaked (we always evaluate on unseen examples), but in the sense that the learner effectively observes more data from the single task than its context length at evaluation. Indeed, when we do not permute input embeddings, we observe task memorization (i.e., data leakage) and the model does very well at evaluation with even close to empty context.
>
> To verify that there is no data leakage when we enable permuting input embeddings (+P), we take one of the “Proposed + P” runs trained on Waterbirds and evaluate it on ICL sequences where input embeddings of each sequence are rotated with a random *rotation matrix*. As the set of permutation matrices is a measure-zero subset of general rotation matrices, we expect that in case of data leakage we would observe degraded performance, as the model would be expecting randomly permuted embeddings of some memorized embedding space. In results presented in Figure 19 of the updated draft, we see that under this new evaluation the results are the same (up to statistical noise), failing to confirm that there is any data leakage when input embeddings are permuted during training.
>
> > Suggestion 1. Provide a theoretical foundation: Develop a theoretical framework that explains why the proposed methods are expected to mitigate spurious correlations. This could involve analyzing the underlying mechanisms through which these methods operate and providing formal guarantees or bounds on their performance.
>
> Regarding theory, while we agree on the usefulness of theoretical analysis, we believe it can come after our experimental work.

---

> > ### Comment · Reviewer_ZBp4 · 2024-11-30
> >
> > I thank the authors for the response. I believe my concern about data leakage has been addressed, but my concern about the oversimplification of the experimental model and the somewhat toy-like experimental setup still remains. As I mentioned, this paper lacks a strong theoretical foundation for these methods, and the experimental validation is somewhat restricted. I believe that when the main contribution of the paper comes from the experiments, comprehensive and realistic experimental settings are important, which this paper still lacks.

---

### Official Review · Reviewer_C43K · 2024-11-04

**Soundness:** 2
**Presentation:** 3
**Contribution:** 3
**Rating:** 5
**Confidence:** 2

**Summary:**

In this paper, authors explored the limit of current ICL framework by present more challenging setup in image classification with presence of spurious correlation in the features. They showed that existing approaches can lead to poor performance under distribution shift due to memorization issue. They proposed a new way of forming new ICL instances that can outperform baselines and mitigate this issue.

**Strengths:**

1. This paper proposes an interesting idea and the method is novel.
2. Conducted analysis and experiments to demonstrate the idea and effectiveness.
3. Overall, the paper is well-written.

**Weaknesses:**

1. Impact of the paper is limited given current setup and experiments.
2. Experiments are very limited and need more benchmarks to show this approach is generalizable to other cases.

**Questions:**

1. Some simulations and theoretical analysis or proof can make the paper stronger.
2. It might worth to explore this idea with more benchmarks and experiments in visionLLM or LLM setup? ICL are frequently used in those models with prompts and I am curious if this method can improve the capability of ICL there, which can greatly enhance the impacts.

---

> ### Author Response · Authors · 2024-11-24
>
> We thank the reviewer for their thoughtful comments. We appreciate that the reviewer finds our work interesting, novel and well-written. We provide responses to the reviewer’s specific questions below.
>
> > 1. Impact of the paper is limited given current setup and experiments.
>
> We believe our work can have a significant impact for multiple reasons.
> * To our best knowledge, this is the first work where it is shown to a degree that one could **learn** a learning algorithm that is robust to spurious correlations instead of manually designing algorithms, which has been the main approach so far. In fact, in many cases, the learning algorithm outperforms GroupDRO, which is a strong baseline. This contribution is similar in spirit to the impactful contribution of Garg et al. (2022), where they demonstrate that in no distribution shift settings, one can learn classification or regression algorithms instead of manually designing them.
> * While the proposed technique might not be the best approach for learning a robust algorithm, we believe some of the techniques we explored can be very useful in future iterations. For example, the permuting input embeddings (the +P technique) can be seen as an independent contribution that allows to control in-context vs in-weights learning, for which not many techniques exist [1, 2]. Similarly, introducing intermediate queries (Figure 1b) is a novel idea, which can be picked up by the community and improved upon.
> * We also show that there is utility in constructing and training on datasets with synthetic spurious features. In our preliminary experiments, we explored a few ways of doing this and selected the grafting approach. Nevertheless, our findings can spark further research in constructing diverse and challenging ICL instances, which can then be added to the pretraining mixtures of modern multimodal LLMs to enhance their robust in-context learning abilities.
> * More generally, our findings build further evidence that standard in-context learning is susceptible to spurious correlations and distribution shifts, and new methods are required to learn more robust algorithms.
>
> > 2. Experiments are very limited and need more benchmarks to show this approach is generalizable to other cases.
>
> We are conducting experiments on a new dataset and will update if the results become available on time. Additionally, in response to Reviewer ZBp4, we have conducted new experiments on Celeb-A with a larger network and observed qualitatively similar results. Please see Appendix B of the updated draft.
>
> > Comment 1. Some simulations and theoretical analysis or proof can make the paper stronger.
>
> Regarding simulations, could you please clarify what simulations would be helpful? Regarding theory, while we agree on the usefulness of theoretical analysis, we believe it can come after our experimental work.
>
> > Comment 2. It might worth to explore this idea with more benchmarks and experiments in visionLLM or LLM setup? ICL are frequently used in those models with prompts and I am curious if this method can improve the capability of ICL there, which can greatly enhance the impacts.
>
> In this work, we explore whether one could use in-context learning to learn new algorithms that are robust to spurious features. Within this context and setup, we cannot do experiments with multimodal/vision LLMs. However, there is one way which we envision our work could benefit future generations of multimodal LLMs. Namely, the diverse synthetic in-context learning instances designed in Section 3, could be added to pretraining mixtures of multimodal LLMs to enhance their robust ICL abilities.
>
>
>
> ### **References**
>
> [1] Garg S, Tsipras D, Liang PS, Valiant G. What can transformers learn in-context? a case study of simple function classes. NeurIPS 2022.
>
> [2] Chan S, Santoro A, Lampinen A, Wang J, Singh A, Richemond P, McClelland J, Hill F. Data distributional properties drive emergent in-context learning in transformers. NeurIPS 2022.
>
> [3] Singh A, Chan S, Moskovitz T, Grant E, Saxe A, Hill F. The transient nature of emergent in-context learning in transformers. NeurIPS 2024.

---

### Official Review · Reviewer_RMDA · 2024-11-08

**Soundness:** 3
**Presentation:** 2
**Contribution:** 3
**Rating:** 6
**Confidence:** 3

**Summary:**

This paper investigates the problem of in-context learning in the presence of spurious correlations. The authors first examine the single-task setting and demonstrate that conventional approaches are vulnerable to spurious correlations. To address this issue, they propose several techniques to improve generalization in the presence of spurious correlations for in-context learning. However, they find that these methods do not generalize well to multiple tasks. Consequently, they develop additional techniques for training on diverse datasets and demonstrate the effectiveness of their method through experiments.

**Strengths:**

1. The paper addresses an important and interesting problem in in-context learning, as spurious correlations are a common issue in real-world scenarios.
The proposed method is generally sound and is validated by experimental results.

**Weaknesses:**

1. A major issue, although acknowledged in Section 5, is the requirement of spurious feature annotations. This is a strong assumption in practice and may significantly limit the applicability of the proposed method.
2. In Section 2.2, an intuitive method would be to sample all in-context samples $x_i$ such that they all have a uniform group distribution. This could mitigate the issue of spurious correlations. Could the authors explain why they did not adopt this method? What are the experimental results of this approach?
3. The paper would benefit from more intuitive figures to explain the settings and proposed methods. Given the complexity of these settings, visual aids would help readers better understand them.
4. The authors construct a simulated dataset, Waterbird-Severe, to validate the effectiveness of their method. However, this dataset may not be realistic and contains a spurious correlation that is unrealistically strong. As a result, it would be more convincing if the authors conducted experiments on more realistic datasets.

**Questions:**

See the weakness part.

---

> ### Author Response · Authors · 2024-11-24
> **Author response (part 1)**
>
> We thank the reviewer for their careful review and thoughtful comments. We appreciate that the reviewer finds our work generally sound and validated by experimental results, addressing an important and interesting problem. We provide responses to the reviewer’s specific questions below.
>
> > 1. A major issue, although acknowledged in Section 5, is the requirement of spurious feature annotations. This is a strong assumption in practice and may significantly limit the applicability of the proposed method.
>
> Thank you for raising this subtle aspect. We have elaborated on this in the updated draft but address here too in detail.
> * First, it is important to note that we are considering classification problems where the learner is given training data only from *one environment/domain*. Because the learner observes only a single environment, in general, there is no way for it to determine which features are spurious. This implies that in single domain settings, in general, one needs to specify the spurious feature, either implicitly through world knowledge and inductive biases or with explicit annotations.
> * Second, it is helpful to distinguish the two cases: (a) training an in-context learner that does not take spurious annotations (i.e., $\tilde{y}$ represents $y$) and (b) training an in-context learner that does take spurious annotations as input (i.e., $\tilde{y}$ represents group $g$). As stated at lines 150-159, we can think of (a) as algorithms having the same signature as ERM and (b) as algorithms having the same signature as GroupDRO. In training of both ERM-like (a) and GroupDro-like (b) learners, we require spurious annotations, be it a single task (Section 2) or multiple task (Section 3) setting. In the multiple task setting of Section 3, this is not an issue because the tasks are synthetically constructed and by design spurious annotations are available. At test time, when a learned ERM-like (type a) algorithm is applied to a new task, we do not need spurious feature annotations. However, when a learned GroupDro-like (type b) algorithm is applied to a new task, it requires spurious feature annotations, just as GroupDRO does.
>
> > 2. In Section 2.2, an intuitive method would be to sample all in-context samples such that they all have a uniform group distribution. This could mitigate the issue of spurious correlations. Could the authors explain why they did not adopt this method? What are the experimental results of this approach?
>
> We have added clarifications and a new experiment on this in the updated manuscript, but we will detail them here too. Indeed, in the single task setting (Section 2), one can train on contexts with balanced groups. However, ERM-like ($\tilde{y}$ representing $y$) learners trained on balanced contexts would be useless for related tasks with an unannotated spurious feature. This distinction becomes even more important in the multiple task setting (Section 3), where the goal is to learn an algorithm that is useful for new related tasks. Nevertheless, we agree that it is still useful to understand how well this approach does in the single task setting of Section 2. For this end, we trained in-context learners on balanced-group sequences consisting of 128 Waterbirds examples. This way each group is represented with 32 context examples. Note that in our main Waterbirds experiments with 512 context examples but group-imbalanced contexts, the minority groups are represented with even less, 25 examples. As the group-balanced sampling context breaks the correlation between the label and spurious feature, we only consider the naive approach of forming ICL sequences (Figure 2). The results presented in Figure 20 of the updated draft show that, as expected, group-balanced sampling improves worst-group accuracy. The naive approach, which again ignores the context and does tasks memorization, reaches 86.08 $\pm$ 1.87 worst-group accuracy with 128 group-balanced context examples, compared to 84.82 $\pm$ 1.26 worst-group accuracy on 512 *group-imbalanced* context examples (see Table 1)). This positive effect of downsampling has been also observed in standard (not in-context) training settings (Nagarajan et al., 2021; Menon et al., 2021; Idrissi et al., 2022). Furthermore, we again see that the proposed technique of permuting embedding dimensions induces strong in-context learning and reaches 90.44 $\pm$ 1.10 worst-group accuracy with 128 group-balanced context examples. This is just a bit below the 91.95 $\pm$ 1.20 worst-group accuracy we get training "Proposed + P" on 512 *group-imbalanced* context examples (see Table 1).

---

> ### Author Response · Authors · 2024-11-24
> **Author response (part 2)**
>
> > 3. The paper would benefit from more intuitive figures to explain the settings and proposed methods. Given the complexity of these settings, visual aids would help readers better understand them.
>
> We have added few illustrations to the updated draft:
> * [+P] Figure 11 visualizes how input embeddings are permuted for each ICL sequence (denoted with +P).
> * [+G] Figure 12 visualizes how ICL sequences are constructed when example groups are passed as input.
> * [+I] Figure 13 visualizes how ICL sequences are constructed when promotion of induction heads is enabled.
> * [iNaturalist, in response to Review WYsn], Figure 14 visualizes the data preprocessing we do to the iNaturalist dataset.
>
> > 4. The authors construct a simulated dataset, Waterbird-Severe, to validate the effectiveness of their method. However, this dataset may not be realistic and contains a spurious correlation that is unrealistically strong. As a result, it would be more convincing if the authors conducted experiments on more realistic datasets.
>
> In the single task setting of Section 2, we see that Waterbirds-Severe is indeed a hard task, but it can be solved reasonably well with 512 context examples. However in the multiple task setting of Section 3, we indeed observed a failed generalization from iNaturalist to Waterbirds-Severe. In agreement with the reviewer, we hypothesized that Waterbirds-Severe is too challenging and considered less challenging variants by varying the norm of the added spurious vector (please see Figure 10). Additionally, we are conducting experiments on a new dataset and will update if the results become available on time.

---

> > ### Comment · Reviewer_RMDA · 2024-11-27
> >
> > I thank the authors for their response and will keep my score unchanged.

---

### Meta-Review · Area_Chair_Z6FG · 2024-12-29

**Metareview:**

This paper explores how to make in-context learning (ICL) robust against spurious correlations in classification tasks. The authors find that standard ICL approaches are vulnerable to spurious features and tend to memorize tasks rather than truly learn from context. To address this, the authors propose several techniques including permuting input embeddings and constructing ICL sequences with intermediate queries to simulate distribution shifts, which outperform baseline methods like ERM and GroupDRO on specific tasks. They show it's possible to train ICL models that generalize to unseen tasks by training on diverse synthetic data (this isn't novel and has been shown previously, eg. see [1]). The main strengths are the paper's novel techniques for improving ICL robustness and demonstration of both single-task and multi-task generalization capabilities. The reviewers pointed out weaknesses in limited theoretical foundation for why the proposed methods work, using relatively simple experimental settings with a single model architecture and limited datasets, and strong assumptions about having access to spurious feature annotations during training.

[1] https://arxiv.org/abs/2207.04179

**Additional Comments On Reviewer Discussion:**

See above.

---

### Decision · Program_Chairs · 2025-01-22

Reject